# Roles of C/EBP class bZip proteins in the growth and cell competition of *Rp* ('Minute') mutants in *Drosophila*

Jorge Blanco[1†], Jacob C Cooper[2], Nicholas E Baker[1]*

[1]Department of Genetics, Albert Einstein College of Medicine, New York, United States; [2]School of Biological Sciences, University of Utah, Salt Lake City, United States

**Abstract** Reduced copy number of ribosomal protein (*Rp*) genes adversely affects both flies and mammals. *Xrp1* encodes a reportedly *Drosophila*-specific AT-hook, bZIP protein responsible for many of the effects including the elimination of *Rp* mutant cells by competition with wild type cells. Irbp18, an evolutionarily conserved bZIP gene, heterodimerizes with Xrp1 and with another bZip protein, dATF4. We show that Irbp18 is required for the effects of Xrp1, whereas dATF4 does not share the same phenotype, indicating that Xrp1/Irbp18 is the complex active in *Rp* mutant cells, independently of other complexes that share Irbp18. Xrp1 and Irbp18 transcripts and proteins are upregulated in *Rp* mutant cells by auto-regulatory expression that depends on the Xrp1 DNA binding domains and is necessary for cell competition. We show that Xrp1 is conserved beyond *Drosophila*, although under positive selection for rapid evolution, and that at least one human bZip protein can similarly affect *Drosophila* development.

*For correspondence:
nicholas.baker@einstein.yu.edu

Present address: †Department of Ecology, Evolution and Behavior, University of Minnesota, St. Paul, USA

## Introduction

Organ structure and function can be affected by the presence of abnormal cells and there is increasing evidence that they can be removed by both cell intrinsic and cell-nonautonomous mechanisms. A cell intrinsic mechanism is exemplified by P53, which activates a transcriptional program that leads either to cell cycle arrest or to the elimination of the damaged cells through apoptosis (*Kastenhuber and Lowe, 2017*). Abnormal cells can also be removed by cell interactions in the process of cell competition, which recognizes and eliminate cells that are different from their neighbors. The classic example of cell competition is that of the *Minute* (*M*) mutations in *Drosophila*, which are loss of function mutations in ribosomal protein (*Rp*) genes (*Marygold et al., 2007*). Heterozygous *Rp*[+/-] animals are viable, although they generally display a slower cell proliferation rate and developmental delay (*Bridges and Morgan, 1923*; *Morata and Ripoll, 1975*) but *Rp*[+/-] cells undergo apoptosis when surrounded by wild-type *Rp*[+/+] cells (*Morata and Ripoll, 1975*; *Simpson and Morata, 1981*; *Moreno et al., 2002*; *Li and Baker, 2007*). Such non-autonomous cell competition also affects a number of other genotypes of cells in both *Drosophila* and in mammals (*Amoyel and Bach, 2014*; *Clavería and Torres, 2016*; *Di Gregorio et al., 2016*; *Merino et al., 2016*; *Baker, 2017*; *Maruyama and Fujita, 2017*; *Nagata and Igaki, 2018*). Interestingly, P53 is important for some examples of cell competition in mammals, but dispensable for the elimination of *Rp*[+/-] cells in *Drosophila* (*Baker et al., 2019*). Although the potential roles of cell competition in development and in disease such as cancer are of considerable interest, little is yet known about molecular mechanisms of cell competition.

We, and others, identified Xrp1 as a key factor in the cell competition of *Rp*[+/-] cells (*Lee et al., 2016*; *Baillon et al., 2018*; *Lee et al., 2018*). *Xrp1* loss-of-function mutations allow *Rp*[+/-] cells to survive when surrounded by wild-type (*Rp*[+/+]) cells, preventing their elimination by cell

competition (*Baillon et al., 2018*; *Lee et al., 2018*). Remarkably, Xrp1 is also required for the slow growth, reduced translation, and most of the gene expression changes shown by $Rp^{+/-}$ cells, showing that Xrp1 is a central mediator of these effects of *Rp* gene mutations, none of which seems to depend simply on a reduced number of ribosomes (*Lee et al., 2018*). Xrp1 encodes a Basic region Leuzine-Zipper (bZIP) protein that also has an AT-hook domain, and was known earlier as a p53-target that is also implicated in P element transposition (*Brodsky et al., 2004*; *Akdemir et al., 2007*; *Francis et al., 2016*). Recently it has also been implicated in coordination of organ growth following local growth retardation (*Boulan et al., 2019*). bZip proteins typically bind DNA as homo- or heterodimers and many are evolutionarily conserved (*Amoutzias et al., 2007*; *Reinke et al., 2013*). Dimerization of *Drosophila* bZIP proteins has been analyzed in silico and in vitro (*Fassler et al., 2002*; *Reinke et al., 2013*). The bZIP protein encoded by the *CG6272* gene was the only heterodimer partner of Xrp1 identified by in vitro FRET assays (*Reinke et al., 2013*). This heterodimer is also the sequence-specific DNA-binding component of a multiprotein complex that binds to the P-element Terminal Inverted Repeats leading to the naming of CG6272 as Inverted Repeat Binding Protein 18 (IRBP18) (*Francis et al., 2016*). Unusually, *Xrp1* has been described as specific to the genus *Drosophila* (*Akdemir et al., 2007*), whereas *irbp18* is well-conserved and belongs to the CAAT/Enhancer Binding Protein (C/EBP) superfamily of transcription factors, being most similar to human C/EBPγ (*Ramji and Foka, 2002*; *Francis et al., 2016*). IRBP18 can also heterodimerize with a second bZIP protein, dATF4 (*Reinke et al., 2013*). dATF4, encoded by the *cryptocephal* (*crc*) gene, controls molting and metamorphosis as a cofactor for the ecdysone receptor (*Gauthier et al., 2012*). Ecdysone is an important signal for imaginal disc growth (*Delanoue et al., 2010*; *Jaszczak and Halme, 2016*). In addition, dATF4 is neuroprotective in a model of Parkinsons Disease (*Celardo et al., 2017*), and regulates 4E-BP transcription downstream of GCN2 during nutrient stress (*Kang et al., 2017*; *Malzer et al., 2018*). dATF4 can heterodimerize with a further C/EBP protein, encoded by *slow border cells* (*slbo*) (*Reinke et al., 2013*). Thus, Xrp1 is potentially connected to a network of more conserved C/EBP-class bZip proteins, through IRBP18 (*Figure 1A*).

In the present study, we have analyzed the role of other bZip proteins connected to Xrp1 in *Rp* mutant cell competition. Out of the many possibilities, we were particularly interested in whether *Xrp1* acts positively in cell competition as a heterodimer with IRBP18, potentially a *Drosophila*-specific complex, or behaves as a competitive inhibitor of the conserved stress response protein dATF4/Crc by sequestering IRBP18 away from it. Our results show that *Xrp1* and *irbp18* are both necessary for the slow-growth and cell competition of $Rp^{+/-}$ cells, apparently independently of other IRBP18 partners, consistent with the first hypothesis. Additionally, we find that the upregulation of both *Xrp1* and *irbp18* in $Rp^{+/-}$ cells depends on one another's function, and that the DNA binding and dimerization domains are important for the function and regulation of these genes, further evidence of an auto-regulatory loop based on the Xrp1/IRBP18 heterodimer and necessary for cell competition. Finally, we identify rapidly-evolving *Xrp1* homologs beyond *Drosophila* and suggest that divergent mammalian proteins might resemble Xrp1 functionally.

## Results

### Irbp18 is required for cell competition

Cell competition eliminates $Rp^{+/-}$ cells from the developing wing mosaics in an Xrp1-dependent manner (*Lee et al., 2016*; *Baillon et al., 2018*; *Lee et al., 2018*). Two distinct assays were used here to analyze the requirement for Irbp18 in cell competition, making use of the *irbp18^f05006* mutation, a PiggyBac insertion within the *irbp18* open reading frame that eliminates IRBP18 protein production (*Francis et al., 2016*). In the first assay, mitotic recombination in wild-type ($RpL36^{+/+}$) progenitor cells generates clones of $RpL36^{+/-}$ cells, which are normally eliminated by cell competition (*Figure 1B*) (*Tyler et al., 2007*). By contrast, in the i*rbp18^f05006/f05006* background the $RpL36^{+/-}$ clones were always recovered, indicating a failure of cell competition when *irbp18* was mutated (*Figure 1C*). In the second assay, mitotic recombination in $RpS18^{+/-}$ progenitor cells generated wild-type clones ($RpS18^{+/+}$) that progressively eliminated their slower-growing neighbors and eventually occupied almost the entire wing imaginal disc (86% on average. *Figure 1D,F*). By contrast, mutating *irbp18* prevented the $RpS18^{+/+}$ cells from taking over the $RpS18^{+/-}$ wing imaginal disc, where they occupied a much smaller fraction (25%, on average) (*Figure 1E,F*). Remarkably, $RpS18^{-/-}$ cells,

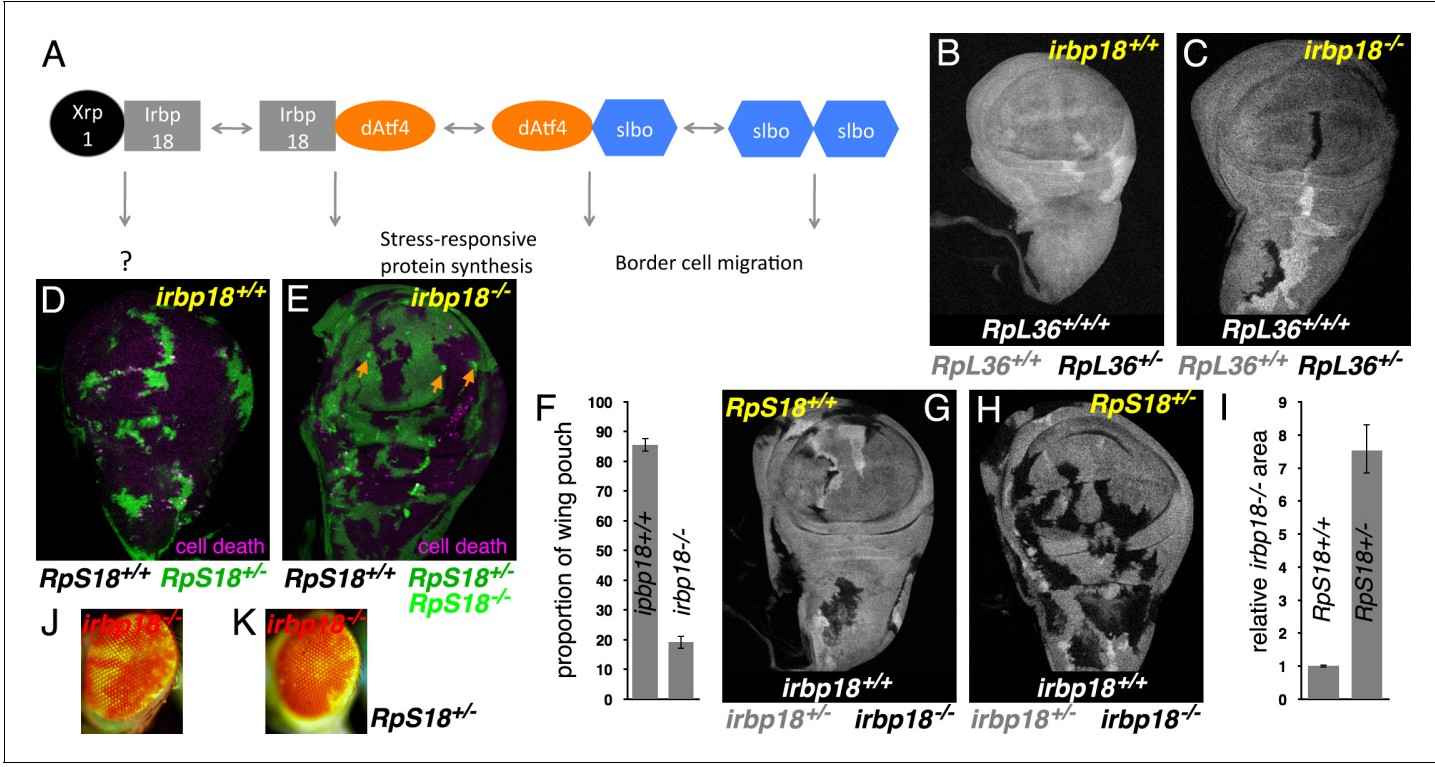

**Figure 1.** Xrp1 and related bZip proteins in cell competition. (**A**) Known dimers of *Drosophila* C/EBP –class bZip proteins and their potential functions. (**B,C**) Mitotic recombination in *RpL36*[+/+] wing discs (grey) generates clones of *RpL36*[+/+/+] cells (light grey) and reciprocal clones of *RpL36*[+/-] cells (black, lacking beta-Gal labeling). *RpL36*[+/-] clones that did not survive in the *irbp18*[+/+] background (**B**) always survived in the *irbp18*[-/-] background (**C**). (**D,E**) Mitotic recombination in *RpS18*[+/-] wing discs (green) generates clones of *RpS18*[+/+] cells (black,)lacking GFP expression. In the *irbp18*[+/+] background (**D**) these have a growth and competitive advantage and come to dominate wing disc territory, eliminating remaining *RpS18*[+/-] cells by cell death (anti-active caspase DCP1 labeling in magenta). *RpS18*[+/+] cells had less advantage in the *irbp18*[-/-] background (**E**). In addition, reciprocally recombinant *RpS18*[-/-] cells survived as small clones (bright green, eg arrows in E). (**F**) Quantitative comparison of *RpS18*[+/+] clone growth in *irbp18*[+/+] and *irbp18*[-/-] backgrounds. Wing pouch areas were the same in *irbp18*[+/+] and *irbp18*[-/-] backgrounds (p=0.191, two-tailed t-test), but the *RpS18*[+/+] fractions were not (p<0.0001, two-tailed t-test). Data derived from measurements of 4 *irbp18*[+/+] discs and five *irbp18*[-/-] discs. (**G,H**) Mitotic recombination of the *irbp18* locus in the *RpS18*[+/+] and *RpS18*[+/-] backgrounds. Reciprocal clones of *RpS18*[+/+] *irbp18*[+/+] and *RpS18*[+/+] *irbp18*[-/-] cells grew comparably (**G**) whereas clones of *RpS18*[+/-]*irbp18*[-/-] cells expanded at the expense of *RpS18*[+/-]*irbp18*[+/+] cells and *RpS18*[+/-]*irbp18*[+/+] cells (**H**). (**I**) Quantification of growth in *RpS18*[+/-] and *RpS18*[+/+] wing discs (eg panels G,H). *irbp18*[-/-] clone area was compared to reciprocal *irbp18*[+/+] controls. Probability that clone areas are the same in the *RpS18*[+/+] and *RpS18*[+/-] backgrounds = $1.4 \times 10^{-9}$ (2-tailed t-test with unequal variances). Data derived from measurements of 7 *RpS18*[+/+] discs and 9 *RpS18*[+/-] discs. (**J**) *irbp18*[-/-] clones (pigmented) contributing to the adult eye. (**K**) *irbp18*[-/-] clones (pigmented) occupy nearly all the *RpS18*[+/-] eye. Genotypes B) hsFlp/M(1)Bld; P[RpL36+ w+] arm-LacZ FRT80B/FRT80B. C) hsFlp/M(1)Bld; P[RpL36+ w+] arm-LacZ FRT80B/irbp18[f05006] FRT80B. D) y w hs-FLP; FRT42D ubi-GFP M( f05006)56F/FRT42D. E) y w hs-FLP; FRT42D ubi-GFP M(2)56F/FRT42D; irbp18[f05006]/irbp18[f05006]. G) y w hs-FLP; +/+; irbp18[f05006] FRT80B /arm LacZ FRT80B. H) y w hs-FLP; FRT42D ubi-GFP M(2)56F*/+; irbp18[f05006] FRT80B/arm LacZ FRT80B. J) y w ey-FLP/Y; FRT42D ubi-GFP/+; irbp18[f05006] FRT80B/FRT80B K) y w ey-FLP/Y; FRT42D ubi-GFP RpS18/+; irbp18[f05006] FRT80B/FRT80B. Note that the *irbp18*[f05006] allele includes a $w^+$ element that is responsible for most of the eye pigmentation.

The online version of this article includes the following source data for figure 1:

**Source data 1.** Measurements of clone and wing pouch sizes corresponding to *Figure 1F*.
**Source data 2.** Clone size data corresponding to *Figure 1I*.

generated as reciprocal recombinants in the *RpS18*[+/-] wing discs, survived as small clones in the absence of *irbp18*, whereas they normally disappear very quickly in *irbp18*[+/+] backgrounds (*Figure 1E*). It is already known that *Xrp1* mutations similarly protect *RpS18*[-/-] clones (*Lee et al., 2018*). These similar effects of *irbp18* and *Xrp1* mutations are expected if it is the Irbp18/Xrp1 heterodimer that promotes both the competitive elimination of *Rp*[+/-] cells and also the prompt apoptosis of *Rp*[-/-] cells.

To confirm that Irbp18 acts in *Rp*[+/-] cells like Xrp1 does, *irbp18*[f05006] homozygous clones were induced in both wild-type and *Rp*[+/-] genetic backgrounds. In the wild-type, *Rp*[+/+] background,

*irbp18$^{f05006}$* homozygous clones were the same size as their *irbp18$^{+/+}$* controls (**Figure 1G,I**). By contrast, *RpS18$^{+/-}$irbp18$^{-/-}$* clones were significantly larger compared to their *RpS18$^{+/-}$irbp18$^{+/+}$* controls, indicating that Irbp18 acts in *RpS18$^{+/-}$* cells to slow their growth, and may also make them less competitive (**Figure 1H,I**). *RpS18$^{+/-}$Irbp18$^{+/+}$* clones were disadvantaged in eye development as well as in the wing disc (**Figure 1J,K**). These results indicate that whereas Irbp18 had no effect on growth of cells in wild-type wing or eye discs, it reduced the growth and competitiveness of *Rp$^{+/-}$* cells, like Xrp1 does.

## Role of Crc/dATF4 in Minute cells

Irbp18 can also heterodimerize with Crc/dATF4, a protein that plays several roles in growth (**Gauthier et al., 2012**; **Kang et al., 2017**; **Malzer et al., 2018**) (**Figure 1A**). We did not succeed in recombining *crc/dATF4* mutations with FRT42D, to which it is very closely linked, and so used the TIE-DYE system for an RNAi approach rather than mitotic recombination of *crc* mutations. In TIE-DYE, FLP-mediated excision of transcriptional stop sequences independently enables transcription of GFP, b-galactosidase, and Gal4 proteins, which thereafter mark the independent lineages descended from the recombined cells (**Worley et al., 2013**) (**Figure 2A**). We used TIE-DYE in an *RpS18$^{+/-}$* background to express dsRNA specific for Crc/dATF4, *Xrp1*, or *w* along with UAS-His2A::mRFP in recombinant cell clones, and compared their contributions to control clones in the same wing discs. Xrp1 knockdown cell-autonomously enhanced growth of *RpS18$^{+/-}$* cells (**Figure 2B, E**), consistent with previous observations of *Xrp1* mutant alleles (**Lee et al., 2018**), but Crc/dATF4 knockdown had the opposite effect, severely restricting the growth of *RpS18$^{+/-}$* cells (**Figure 2C,E**). The small remaining *RpS18$^{+/-}$* dsRNA$^{Crc/dATF4}$ clones were often fragmented, suggesting they were in the process of apoptosis and/or extrusion from the wing disc (**Figure 2D**). Knock down of neither Xrp1 nor dATF4 significantly affected clone size in the wild type (*RpS18$^{+/+}$*) background (**Figure 2F**), although interestingly some *RpS18$^{+/+}$* dsRNA$^{Crc/dATF4}$ clones showed signs of fragmentation (**Figure 2G,H**). These findings indicated that Crc/dATF4 had the opposite effects on growth of *RpS18$^{+/-}$* cells than did *Xrp1* and *Irbp18*.

## Irbp18 is upregulated in *Rp$^{+/-}$* cells

Since *Irbp18* was required in *Rp$^{+/-}$* cells for their cell competition, and Xrp1 expression is higher in *Rp$^{+/-}$* cells, we compared Irbp18 expression in wild-type and *Rp$^{+/-}$* cells. Indeed, like Xrp1 (**Figure 3A,B**), Irbp18 protein was elevated in *RpS18$^{+/-}$* cells in mosaic wing discs (**Figure 3C,D**). We also found that Irbp18 upregulation in *Rp$^{+/-}$* cells was *Xrp1*-dependent, since clones of *RpS18$^{+/-}$Xrp1$^{-/-}$* cells had lower Irbp18 levels (**Figure 3E,F**). Interestingly, clones of *RpS18$^{+/-}$ irbp$^{-/-}$* cells similarly failed to upregulate Xrp1 protein (**Figure 3G,H**). Consistent with these protein findings, mRNA-Seq data (**Lee et al., 2018**) show *irbp18* transcripts elevated more than three-fold in non-mosaic *RpS3$^{+/-}$* and *RpS17$^{+/-}$* wing discs, in an *Xrp1*-dependent manner, much as was also seen for Xrp1 mRNA itself (**Figure 3I,J**). This cross-regulation between *Xrp1* and *irbp18* is simply explained if the Xrp1/Irbp18 heterodimer stimulates transcription of *Xrp1* and/or *irbp18*. Consistent with this, transcription of a *LacZ* enhancer trap construct inserted onto the *Xrp1* locus was detected upon ectopic expression of *Xrp1* (**Figure 3K,L**). Transcriptional cross-regulation does not rule out additional effects of Xrp1 and Irbp18 proteins on one another's stability, which would also affect protein levels.

## *Xrp1* DNA-binding domains necessary for cell competition and auto-regulation

To assess the contribution of the Xrp1 DNA binding domains in loss-of-function studies, we modified the endogenous *Xrp1* locus to generate new HA-tagged alleles lacking individual domains. This was achieved using homologous recombination to first generate an insertion of pTVcherry into the last intron of the *Xrp1* locus (**Figure 4A**). The insertion contains an *attP* sequence and a mini-white gene flanked by *LoxP* sites. This insertion behaves as a loss of function allele of *Xrp1* (*Xrp1$^{attP\ flox}$*; **Figure 4—figure supplement 1A-D**) Cre-mediated excision of the mini-white gene and other sequences leaves only the *attP* site and a single *LoxP*; this allele (*Xrp1$^{attP\ loxP}$*) behaves as a wild type allele of Xrp1 (**Figure 4A**, **Figure 4—figure supplement 1E,F**). Subsequent PhiC31-mediated insertion into the *attP* site of *Xrp1$^{attP\ loxP}$* inserted recombinant exons, leading to synthesis of recombinant Xrp1 proteins encoded at the endogenous locus. An HA-tagged final exon with otherwise wild type

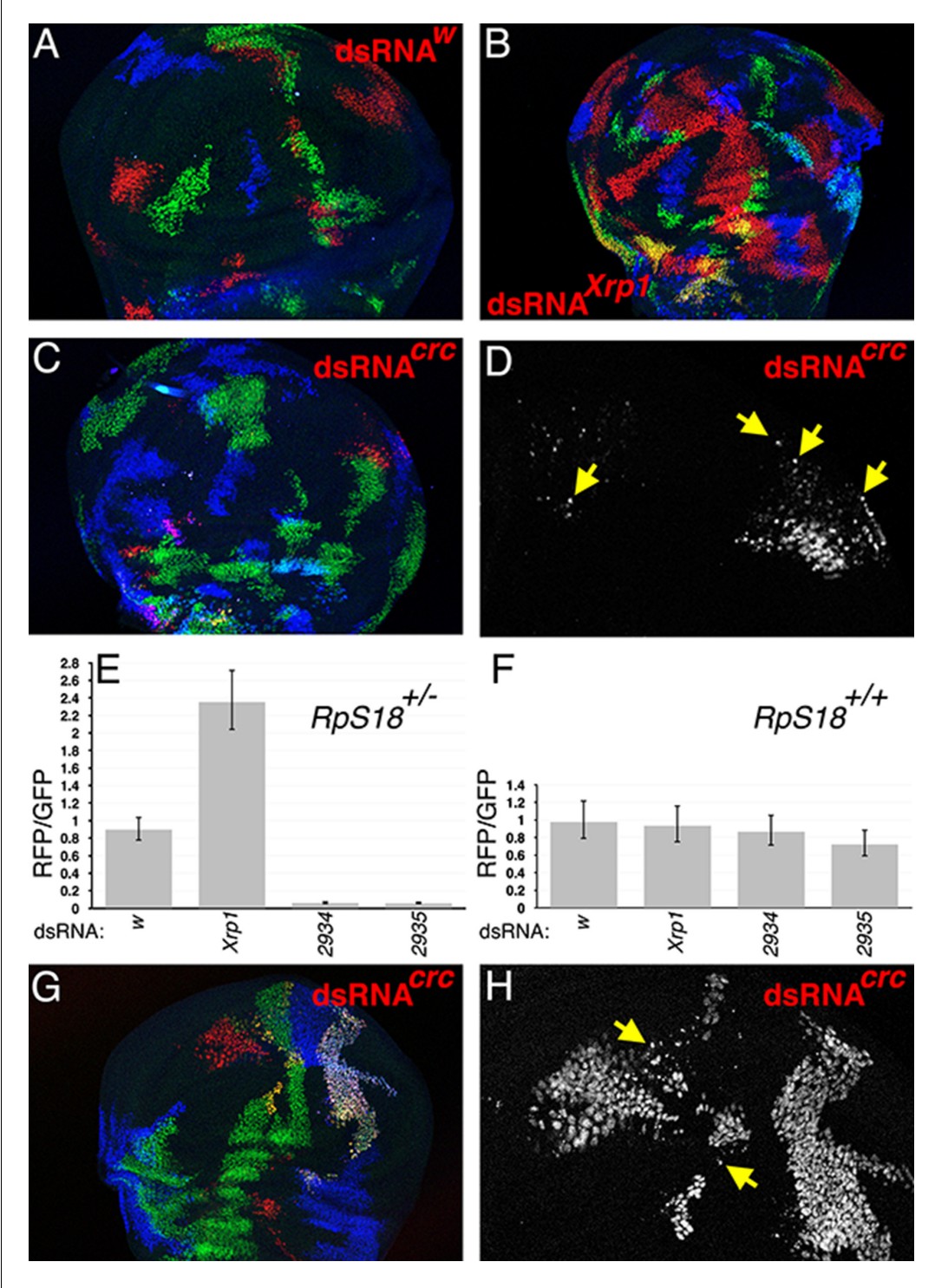

**Figure 2.** dATF4/crc requirement in wild type and $Rp^{+/-}$ wing discs. (**A**) In the TIE-DYE method, independent recombination events generate parallel clones expressing GFP (green), Gal4 (detected through UAS-RFP expression in red), and b-Gal (blue). These parallel clones grow equivalently in $RpS18^{+/-}$ wing discs. In this control, dsRNA targeting transcripts from the *white* gene was co-expressed with RFP under Gal4 control. (**B**) Co-expression of dsRNA targeting *Xrp1* increased the contribution of RFP-positive clones in $RpS18^{+/-}$ wing discs. (**C**) Co-expression of dsRNA targeting *dATF4/crc* decreased the contribution of RFP-positive clones in $RpS18^{+/-}$ wing discs. (**D**) Enlarged portion of the wing disc from panel C to show the fragmentation of RFP-positive, dATF4/crc knock-down $RpS18^{+/-}$ cells (eg arrows). (**E**) Quantification of growth for various dsRNA-expressing RFP-positive clones in $RpS18^{+/-}$ wing discs. Shown is the mean ratio of RFP-positive to GFP-positive areas for each wing disc. Error bars represent ± 1 SEM. Probabilities that clone sizes are the same as for the *w* control (2-tailed t-tests): *Xrp1* - $3.4 \times 10^{-5}$; *2934*–$3.1 \times 10^{-12}$; *2935*–$2.2 \times 10^{-11}$. VDRC2934 and VDRC2935 encode

*Figure 2 continued on next page*

Figure 2 continued

independent, previously-validated dsRNAs targeting dATF4/Crc (**Kang et al., 2017**). Number of wing discs analyzed = 25 (w), 23 (*Xrp1*), 17 (*2934*), 26 (*2935*). (F) Quantification of growth for various dsRNA-expressing RFP-positive clones in *RpS18*$^{+/+}$ wing discs. Probabilities that clone sizes are the same as for the *w* control (2-tailed t-tests): *Xrp1* – 0.88; *2934*–1.0; *2935*–0.68. Number of wing discs analyzed = 17 (w), 26 (*Xrp1*), 16 (*2934*), 19 (*2935*). (G) An example of dATF4/Crc knock-down in RFP-positive clones in a *RpS18*$^{+/+}$ wing disc. (H) Enlarged portion of the wing disc shown in G illustrating that, although substantial RFP-positive territories are present, some of these cells are fragmented (eg arrows). Genotypes A) *y w hs-FLP*; *Act < stop < lacZ-NLS Ubi < stop < eGFP-NLS M(2)56 F/ +*; *Act < stop < GAL4 UAS-His2A::mRFP/P{TRiP.HMS00017}attP2.* B) *y w hs-FLP*; *Act < stop < lacZ-NLS Ubi < stop < eGFP-NLS M(2)56 F/ +*; *Act < stop < GAL4 UAS-His2A::mRFP/P{TRiP.HMS00053}attP2.* C,D) *y w hs-FLP*; *Act < stop < lacZ-NLS Ubi < stop < eGFP-NLS M(2)56 F/ +*; *Act < stop < GAL4 UAS-His2A::mRFP/UAS-dsRNA(Crc)*$^{VDRC2935}$. G,H) *y w hs-FLP*; *Act < stop < lacZ-NLS Ubi < stop < eGFP-NLS / +*; *Act < stop < GAL4 UAS-His2A::mRFP/UAS-dsRNA(Crc)*$^{VDRC2935}$.

The online version of this article includes the following source data for figure 2:

**Source data 1.** Source data for the graph shown in *Figure 2E*.
**Source data 2.** Source data for the graph shown in *Figure 2F*.

sequence was inserted using this strategy (**Figure 4A**). This *Xrp1*$^{HA}$ allele behaved as a *wt* allele during cell competition. That is, *RpS18*$^{+/-}$ cells were normally outcompeted by wild type cells in the homozygous *Xrp1*$^{HA}$ background, occupying only the minority of the wing disc and exhibiting competitive cell death at boundaries with *RpS18*$^{+/+}$ cells (**Figure 4B**). As is seen for the endogenous protein, expression of the Xrp1$^{HA}$ protein was elevated in *RpS18*$^{+/-}$ cells and could be detected with either anti-Xrp1 antibody or with anti-HA (**Figure 4C,D**). In contrast to Xrp1$^{HA}$, alleles encoding Xrp1 proteins lacking either the AT hook or the basic sequence (*Xrp1*$^{ΔAT-HA}$ and *Xrp1*$^{ΔBS-HA}$, respectively) behaved as loss-of-function alleles, allowing *RpS18*$^{+/-}$ cells to occupy most of mosaic discs without experiencing competitive cell death (**Figure 4E,H**). Significantly, both *Xrp1*$^{ΔAT-HA}$ and *Xrp1*$^{ΔBS-HA}$ also prolonged the survival of homozygous *RpS18*$^{-/-}$ clones, which is a feature of *Xrp1*$^{-/-}$ genotypes (**Figure 4E,H**). Both *Xrp1*$^{ΔAT-HA}$ and *Xrp1*$^{ΔBS-HA}$ also impaired the upregulation of *Xrp1* in *RpS18*$^{+/-}$ cells, although Xrp1 protein levels were still elevated in *RpS18*$^{-/-}$ clones (**Figure 4F,G,I,J**)). These results indicate that DNA-binding domains are essential components of *Xrp1* function and auto-regulation in *RpS18*$^{+/-}$ cells. *Xrp1* appears to be elevated in *RpS18*$^{-/-}$ clones by a mechanism independent of auto-regulation, although *Xrp1* function to eliminate *RpS18*$^{-/-}$ cells still depends on the DNA-binding domains.

## *Irbp18* is required for ectopic *Xrp1* effects in wings and eyes

The requirement for DNA binding domains is consistent with transcriptional regulation of cell competition targets. It might have been possible, however, that the *Xrp1* and *Irbp18* genes themselves were the only significant transcriptional targets, and that Xrp1 and Irbp18 caused cell competition through non-transcriptional mechanisms after reaching sufficient expression levels. To explore this possibility, we examined ectopic Xrp1 phenotypes in which Xrp1 transcription was driven independently of its own regulatory sequences. First we tested whether effects of ectopic Xrp1 were also dependent on Irbp18, as expected if Xrp1 functions as a heterodimer with Irbp18 when ectopically expressed.

*Xrp1* over-expression has a strong effect on cell survival in the normal wing imaginal disc, causing the elimination of Xrp1-expressing cell clones within 72 hr (**Figure 5A–B**) (**Tsurui-Nishimura et al., 2013**). By contrast, MARCM clones overexpressing Xrp1 and also homozygous mutant for *Irbp18* (*CG6272*$^{f05006}$) survived, proliferated and were present in the wing epithelium 72 hr after clone induction (**Figure 5C,D**). Therefore the negative effect of *Xrp1* on growth and survival of wing disc clones depended on *Irbp18*. In contrast to *Xrp1*, ectopic expression of *Irbp18* had no effect on clone survival (**Figure 5E,F**).

During eye development, ectopic expression of *Xrp1* using the *GMR-gal4* driver at 18°C, leads to a small, glossy eye (**Figure 6A**). Like the effect on Xrp1 in the wing, the eye phenotype of ectopic Xrp1 was almost completely suppressed in the absence of *irbp18*, so that *irbp18*$^{-/-}$ GMR >Xrp1 eyes were close to wild type in size (**Figure 6B**). This indicates that Irbp18 is required for ectopic Xrp1 activity in the eye also.

Results were slightly different at a higher temperature. At 25°C, GMR > Xrp1 reduces the eye to only a small remnant (**Figure 6C**) (**Tsurui-Nishimura et al., 2013**). Although eye size was rescued

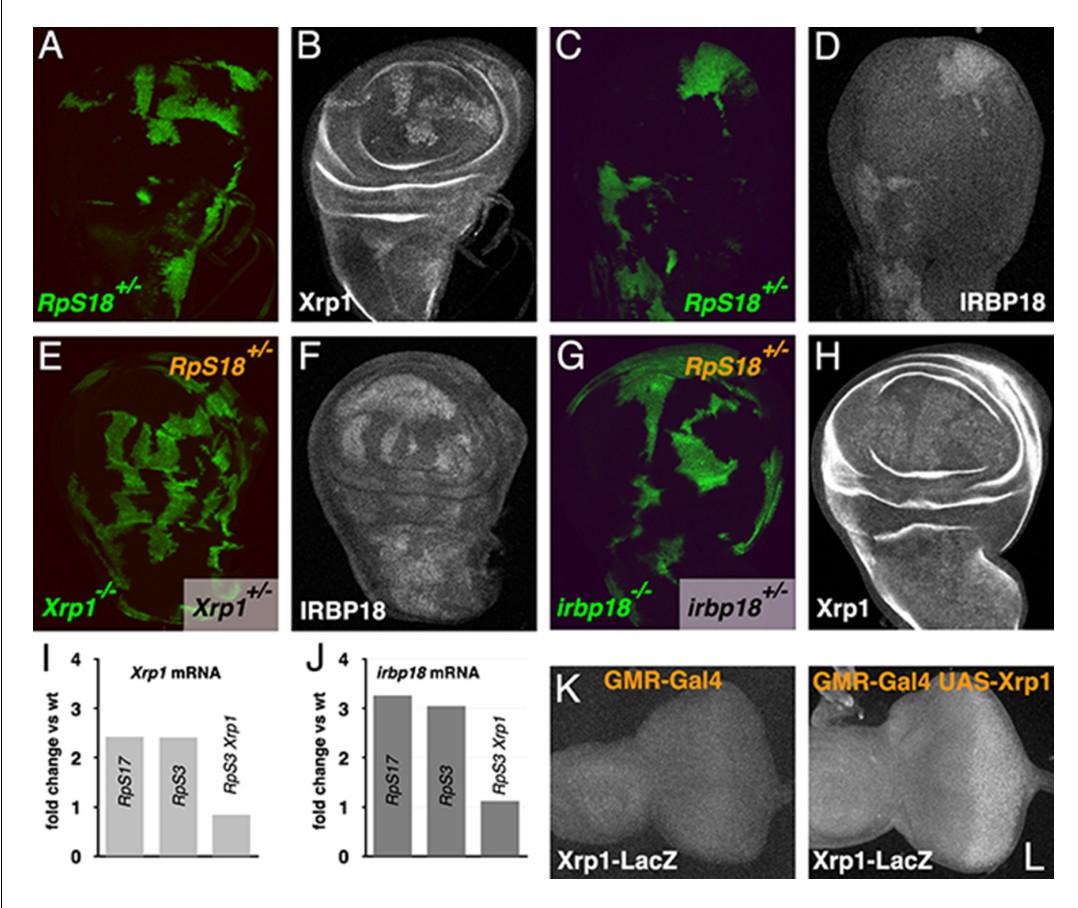

**Figure 3.** Cross-regulation of Xrp1 and IRBP18 expression. (**A,B**) Xrp1 protein (labeled in B) is elevated in $RpS18^{+/-}$ cells (green in A) compared to $RpS18^{+/+}$ cells (unlabelled in A). (**C,D**) Irbp18 protein (labeled in D) is elevated in $RpS18^{+/-}$ cells (green in C) compared to $RpS18^{+/+}$ cells (unlabelled in C). (**E,F**) Irbp18 protein (labeled in F) is elevated in $RpS18^{+/-}Xrp1^{+/-}$ cells or $RpS18^{+/-}Xrp1^{+/+}$ cells (unlabelled in E) compared to $RpS18^{+/-}Xrp1^{-/-}$ cells (green in E), showing that Irbp18 protein expression is Xrp1-dependent. (**G,H**) Xrp1 protein (labeled in H) is higher in $RpS18^{+/-}irbp18^{+/-}$ and $RpS18^{+/-}irbp18^{+/+}$ cells (both unlabeled in G) than in $RpS18^{+/-}irbp18^{-/-}$ cells (green in G), showing that Xrp1 protein up-regulation is Irbp18-dependent. (**I**) mRNA fold-change in wing imaginal discs of various genotypes in comparison to wild type controls. Both $Xrp1$ and $irbp18$ mRNA levels were elevated in $Rp^{+/-}$ wing discs in an Xrp1-dependent manner. Fold changes compared to wild type were determined using DESEQ2 from three biological replicates (*Lee et al., 2018*). The adjusted probabilities that expression levels differed from the wild type control were $7.95 \times 10^{-153}$ ($Xrp1$ in $RpS17^{+/-}$), $1.21 \times 10^{-146}$ ($Xrp1$ in $RpS3^{+/-}$), 0.0149 ($Xrp1$ in $RpS3^{+/-}Xrp1^{+/-}$), $6.75 \times 10^{-32}$ ($irbp18$ in $RpS17^{+/-}$), $9.72 \times 10^{-34}$ ($irbp18$ in $RpS3^{+/-}$), 0.714 ($irbp18$ in $RpS13^{+/-}RpS3^{+/-}Xrp1^{+/-}$). (**K,L**) Over-expression of Xrp1 in the posterior eye disc under control of GMR-Gal4 (**L**) upregulates the Xrp1-LacZ enhancer trap compared to the GMR-Gal4 control (**K**), confirming transcriptional auto-regulation. Genotypes A-D) $y\ w\ hs\text{-}FLP;\ FRT42D\ ubi\text{-}GFP\ M(2)56F/FRT42$. E,F) $y\ w\ hs\text{-}FLP;\ tubP\text{-}GAL4\ UAS\text{-}mCD8::GFP/FRT42D\ ubi\text{-}GFP\ M(2)56F;\ FRT82B\ Xrp1^{M2\text{-}73}/FRT82B\ tubP\text{-}GAL80$. G,H) $y\ w\ hs\text{-}FLP;\ tubP\text{-}GAL4\ UAS\text{-}mCD8::GFP/FRT42D\ ubi\text{-}GFP\ M(2)56\ f^*;\ irbp18^{f05006}\ FRT80B/tubP\text{-}GAL80\ FRT80B$. K) $GMR\text{-}Gal4/+;\ Xrp1^{02515}/+$. L) $GMR\text{-}Gal4/UAS\text{-}Xrp1;\ Xrp1^{02515}/+$.

considerably in the *Irbp18* mutant background, the eyes were not normal, instead having a glossy appearance (*Figure 6D*). This phenotype was definitely due to ectopic Xrp1, because both eye size and appearance were rescued by co-expressing dsRNA for Xrp1 along with ectopic Xrp1 (dsRNA for Xrp1 had no effect on eye development by itself)(*Figure 6E*). The restoration of eye size definitely reflected a requirement for Irbp18, because a less complete rescue was seen when co-expressing dsRNA for Irbp18 (*Figure 6F*), and because restoring Irbp18 expression to the *Irbp18* mutant restored the full GMR > Xrp1 size reduction (*Figure 6G*). Therefore, Xrp1 function at higher temperature remains substantially dependent on Irbp18 but also exhibited an Irbp18-independent component. Ectopic expression of Irbp18 had no effect on the eye (*Figure 6H*), although we confirmed by in situ hybridization that *GMR-gal4* drove strong expression in the eye imaginal disc (*Figure 6I*).

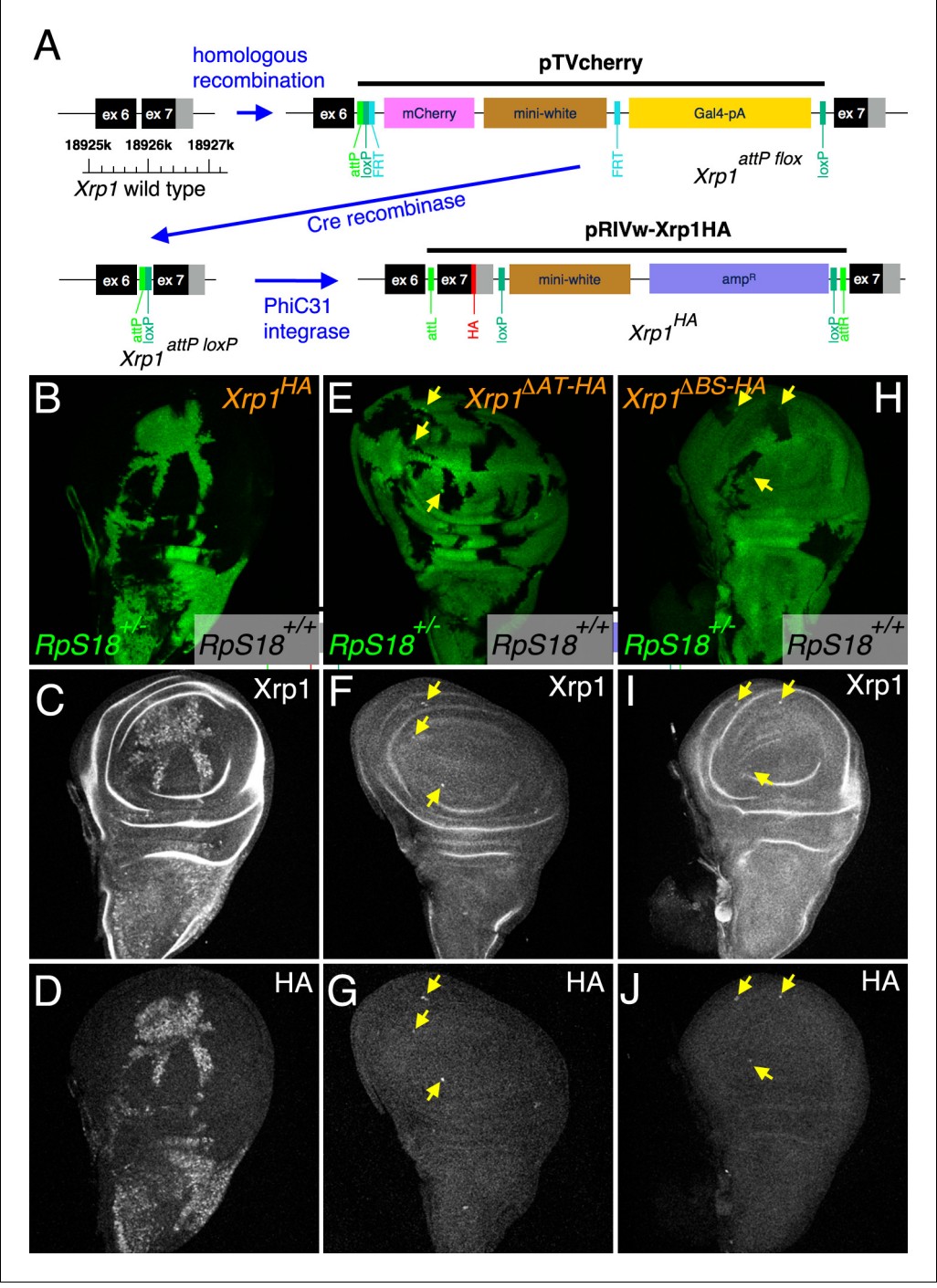

**Figure 4.** Modifications of the endogenous *Xrp1* locus and their consequences for cell competition. (**A**) Sequential modifications of the *Xrp1* locus in the 18925–18926 kb region of chromosome arm 3R by homologous recombination, Cre recombination and PhiC31 integration introduced modified and HA-tagged exon seven sequences into intron 6. (**B–D**) Cell competition in the homozygous *Xrp1^{HA}* background. *RpS18^{+/+} Xrp1^{HA/HA}* clones (unlabeled in B) grow to occupy most of the wing disc at the expense of *RpS18^{+/-}Xrp1^{HA/HA}* cells (green in B), similar to what is seen in the *Xrp1^{+/+}* background (see ***Figure 3A and C***). (**C**) Xrp1 protein is elevated in *RpS18^{+/-}Xrp1^{HA/HA}* cells, as was seen for *RpS18^{+/-}Xrp1^{+/+}* cells (see ***Figure 3B***). (**D**) HA-tagged protein is detected only in the *RpS18^{+/-}Xrp1^{HA/HA}* cells. (**E–G**) Cell competition in the homozygous *Xrp1^{ΔAT-HA}* background. *RpS18^{+/+} Xrp1^{ΔAT-HA/ΔAT-HA}* clones (unlabeled in E) occupy less of the wing disc than *RpS18^{+/-}Xrp1^{ΔAT-HA/ΔAT-HA}* cells (green in E), similar to what is seen in the *Xrp1^{-/-}* or *irbp18^{-/-}* backgrounds (see ***Figure 1E***). Note the survival of small clones of *RpS18^{-/-}Xrp1^{ΔAT-HA/ΔAT-HA}* (brighter green in E, eg arrows). Survival of *Rp^{-/-}* genotypes is a feature of *Xrp1*

*Figure 4 continued on next page*

*Figure 4 continued*

mutants. (F) Xrp1 protein is not elevated in $RpS18^{+/-}Xrp1^{\Delta AT-HA/\Delta AT-HA}$ cells, but is in $RpS18^{-/-}Xrp1^{\Delta AT-HA/\Delta AT-HA}$ (eg arrows). (G) As expected, the Xrp1 protein is HA-tagged (eg arrows). (H–J) Cell competition in the homozygous $Xrp1^{\Delta BR-HA}$ background. $RpS18^{+/+}Xrp1^{\Delta BR-HA/\Delta BR-HA}$ cells (green in H) are not out-grown by $RpS18^{+/+} Xrp1^{\Delta BR-HA/\Delta BR-HA}$ cells. Note the survival of small clones of $RpS18^{-/-}Xrp1^{\Delta BR-HA/\Delta BR-HA}$ (brighter green in H, eg arrows), a feature of Xrp1 mutant genotypes. These $RpS18^{-/-}Xrp1^{\Delta BR-HA/\Delta BR-HA}$ clones are the only cells where Xrp1 protein is elevated (eg arrows I) or HA-tag detected (eg arrows J). Genotypes B-D) *w hs-FLP; FRT42D ubi-GFP M(2)56F/ FRT42; Xrp1^{HA}/Xrp1^{HA}*. E–G). *w hs-FLP; FRT42D ubi-GFP M(2)56F/FRT42; Xrp1^{\Delta AT-HA}/Xrp1^{\Delta AT-HA}*. H–J) *w hs-FLP; FRT42D ubi-GFP M(2)56F/FRT42; Xrp1^{\Delta BR-HA}/Xrp1^{\Delta BR-HA}*..

The online version of this article includes the following figure supplement(s) for figure 4:

**Figure supplement 1.** $Xrp1^{attP\ flox}$ is a loss-of-function allele.

The eye imaginal disc from GMR > Xrp1 exhibited apoptosis soon after Xrp1 was expressed (**Figure 6J,K**). This apoptosis was dependent on *irbp18* (**Figure 6L**). This eye disc region overlaps the second mitotic wave of dividing cells, and cell divisions are reduced in GMR > Xrp1 eye imaginal discs (**Figure 6—figure supplement 1A-D**) (**Tsurui-Nishimura et al., 2013**). Co-overexpression of either anti-apoptotic proteins *P35* or *diap1* (**Figure 6M,N**), or depletion of caspases and pro-apoptotic proteins only rescued the *GMR > Xrp1* eye phenotype modestly (**Figure 6—figure supplement 1E-J**). Whereas over-expression of *UAS-Rheb* enlarged wild type eyes (**Figure 6O**) it had no effect on GMR-Xrp1 eyes (**Figure 6P**). By contrast over-expression of *UAS-yki* both enlarged wild type eyes (**Figure 6Q**) and partially rescued the *GMR > Xrp1* eye phenotype (**Figure 6R**). Co-expression of Yki was also able to rescue the viability of wild-type MARCM clones overexpressing *Xrp1* in

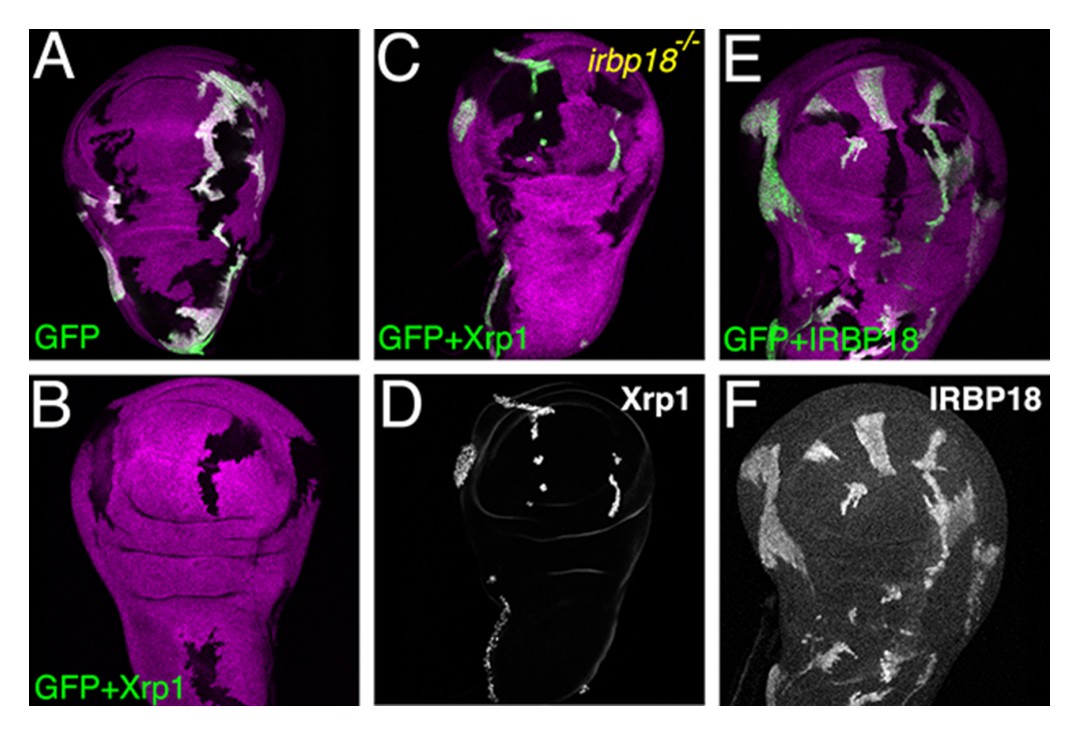

**Figure 5.** Ectopic Xrp1 requires IRBP18 to affect clone survival. (A) Wing imaginal discs containing clones of control cells lacking b-galactosidase (magenta) and reciprocal clones expressing GFP (green). In panels B-D) Xrp1 is co-expressed with GFP. Xrp1 led to complete elimination of GFP-positive lineages in the *irbp18^{+/+}* background (B), but not when *irbp18* was mutated (C). Panel D shows that the GFP-marked clones nevertheless highly over-expressed Xrp1 protein in the *irbp18^{-/-}* background. (E,F) co-expression of Irbp18 with GFP had no effect on clone survival, even though Irbp18 protein was highly over-expressed (F). Genotypes A) *y w hs-FLP; tubP-GAL4 UAS-mCD8::GFP/+; FRT82B arm-LacZ/FRT82B tubP-GAL80*. B) *y w hs-FLP; tubP-GAL4 UAS-mCD8::GFP/UAS-Xrp1; FRT82B arm-LacZ/FRT82B tubP-GAL80*. C–D). *y w hs-FLP; tubP-GAL4 UAS-mCD8::GFP/UAS-Xrp1; irbp18^{f05006} arm-LacZ FRT80/tubP-GAL80 FRT80*. E,F). *y w hs-FLP; tubP-GAL4 UAS-mCD8::GFP/UAS-irbp18; arm-LacZ FRT80/tubP-GAL80 FRT80*.

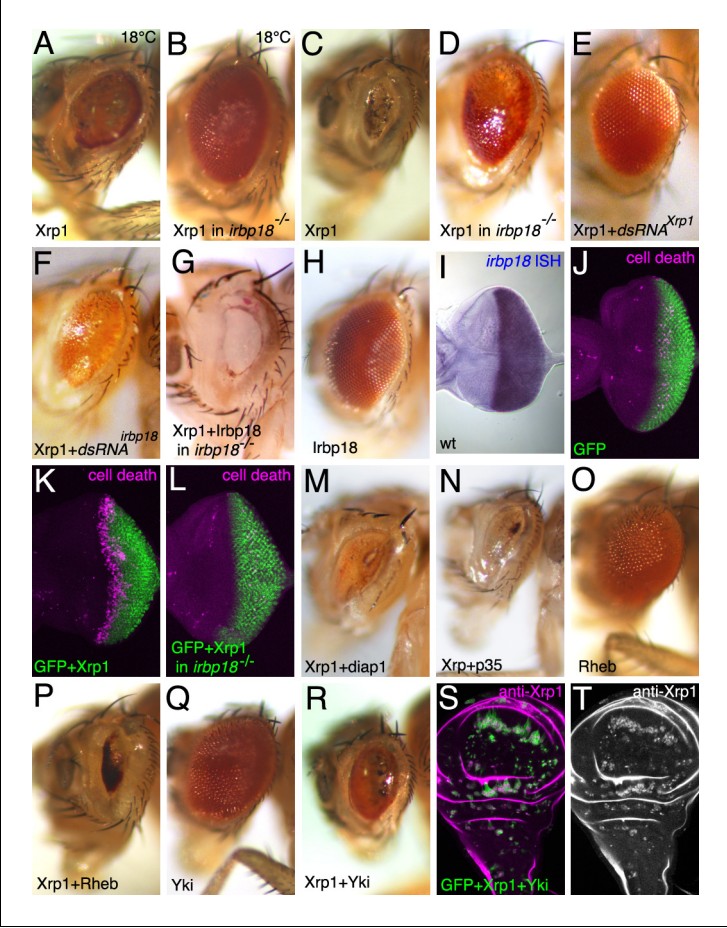

**Figure 6.** Xrp1 over-expression affects the eye in an *irbp18*-dependent manner. Most panels show adult eyes where GMR-Gal4 has been used to express the indicated proteins and dsRNAs posterior to the morphogenetic furrow. Temperature was 25°C except where indicated otherwise. Panels (**J-L**) show eye imaginal discs immunolabeled for mGFP (green) and for cleaved caspase DCP1 (magenta) to reveal apoptosis. Panels S,T show a wing imaginal disc containing clones over-expressing GFP, Xrp1 and Yki. Clones of these cells survived, unlike clones expressing GFP and Xrp1 alone (compare *Figure 5B*), despite expressing highly elevated Xrp1 (panel T). Genotypes A,C,K) GMR-Gal4/UAS-Xrp1; B,D,L) GMR-Gal4/UAS-Xrp1; *irbp18$^{f05006}$/irbp18$^{f05006}$*. E) GMR-Gal4/UAS-Xrp1; UAS-*Xrp1RNAi* /+. F). GMR-Gal4/UAS-Xrp1; UAS-*irbp18RNAi* /+. G) GMR-Gal4/UAS-Xrp1 UAS-*irbp18*; *irbp18$^{f05006/f05006}$*. H,I) GMR-Gal4/UAS-irbp18. J) GMR-Gal4/+. M) GMR-Gal4/UAS-Xrp1; UAS-*diap1*/+. N) GMR-Gal4/UAS-Xrp1; UAS-*p35*/+. O) GMR-Gal4/+; UAS-*Rheb*/+. P) GMR-Gal4/UAS-Xrp1; UAS-*Rheb*/+. Q) GMR-Gal4/+; UAS-*yki*/+. R) GMR-Gal4/UAS-Xrp1; UAS-*yki*/+. S,T) *y w hs-FLP; tubP-GAL4 UAS-mCD8::GFP/UAS-Xrp1 UAS-yki; FRT82B arm-LacZ/FRT82B tubP-GAL80.*.

The online version of this article includes the following figure supplement(s) for figure 6:

**Figure supplement 1.** Cell cycle and genetic interactions in the GMR>Xrp1 genotype.

the wing imaginal disc (*Figure 6S,T*). Thus, as noted previously (*Akdemir et al., 2007*; *Tsurui-Nishimura et al., 2013*), ectopic *Xrp1* affects proliferation and survival, but Yki is able to modulate the effects.

These studies show that Irbp18 remains positively required for Xrp1 function when Xrp1 is transcribed from a transgene, independently of transcriptional auto-regulation. The reason ectopic expression of Irbp18 alone had no effect may be that insufficient Xrp1 is available in normal eye and wing discs to heterodimerize with excess Irbp18.

## *Xrp1* domains necessary for ectopic function

To assess the role of domains required for dimerization and DNA binding, we used UAS-transgenes encoding the Xrp1 protein tagged with HA. We compared full length Xrp1 protein with proteins deleted for the conserved AT-hook, the basic region, or the Leucine Zipper (Figure 8C). As a negative control, we expressed an Xrp1 protein truncated after Tyr546, lacking all of these domains. UAS-Xrp1 is lethal in combination with drivers such as en-Gal4, because ectopic Xrp1 eliminates imaginal disc cells very efficiently, so these analyses were performed using late-acting and tissue-restricted drivers.

These proteins were expressed ectopically in the developing wing using *nub-Gal. Xrp1*$^{1-546}$ expression had almost no effect on the adult wing or wing imaginal disc (*Figure 7A–D*). At 18°C, the *nub-Gal4 UAS-Xrp1*$^{HA}$ genotype died before emergence so that adult wings could not be examined (*Figure 7E*). The third instar wing discs exhibited massive cell death throughout the wing pouch, and complete absence of the developing wing margin structures labeled by the neural transcription factor Senseless (*Figure 7F*). Expression of each of the other deletion proteins were intermediate. They permitted survival, but reduced wing size compared to wild-type controls, showing that they each lacked full Xrp1 function but retained partial function. The weakest phenotype, reflecting least remaining function, was shown by Xrp1$^{ΔBR-HA}$, whose expression only reduced adult wing size modestly, and only in males (*Figure 7G*). Cell death in the developing wing pouch was elevated above background but wing margin structures expressing Senseless were still present (*Figure 7H*). Expression of Xrp1$^{ΔLZ-HA}$ protein led to a larger reduction in wing size, particularly in males, associated with significant cell death in the wing pouch and abnormal differentiation of the wing margin (*Figure 7I, J*). The strongest phenotype, indicating greatest retention of function, was shown by expression of Xrp1$^{ΔAT-HA}$, which led to almost no adults; a single *nub-Gal4 UAS-Xrp1*$^{ΔAT-HA}$ male had underdeveloped, unexpanded wings (*Figure 7K*). Cell death was elevated compared to controls but not to the degree seen after expression of full-length Xrp1 (*Figure 7L*). The phenotypic differences did not reflect differences in protein stability or localization, because similar levels of anti-HA labeling were observed for all the mutant proteins, which was nuclear in all cases (*Figure 7F,H,J,L*).

Similar experiments performed using *GMR-Gal4* to express proteins in the eye also showed that, while deleting the C-terminal region of Xrp1 abolished function (*Figure 7M–P*), deletion of each individual domain reduced function but did not eliminate it (*Figure 7Q–X*). The deleted proteins significantly restored eye size compared to *GMR-Gal4 UAS-Xrp1*, but still had abnormal, glossy eyes. Whereas none of the Xrp1 constructs much affected initial ommatidial patterning, revealed by Senseless labeling, cell death rates were elevated posterior to the morphogenetic furrow, particularly for the full-length Xrp1 protein and the AT hook deletion. As noted previously, eye phenotypes were stronger at 25°C, and at this temperature even *Xrp1*$^{1-546}$ expression modestly affected the eye (*Figure 7—figure supplement 1*).

These data indicate that the C-terminal portion of the Xrp1 protein is necessary overall, and each of the LZ, BR and AT domains contributes to Xrp1 function, even when Xrp1 transcription is under Gal4/UAS control.

## Conservation of Xrp1 beyond *Drosophila*

The *Xrp1* gene has only been reported from the genus *Drosophila* (*Akdemir et al., 2007*). This restricted evolutionary distribution seems surprising if Xrp1 functions with Irbp18, since C/EBP proteins are conserved throughout eukaryotes. This lack of conservation could be due either to lack of selection to retain Xrp1 sequences, leading to evolutionary drift and loss, or to recurrent positive selection of *Xrp1*, causing it to diversify rapidly in different linages over evolutionary time. To test for a signature of recurrent positive selection in the in *Xpr1*, we analyzed sequences of *Xrp1* from 13 species in the Melanogaster group of *Drosophila* using Phylogenetic Analysis by Maximum Likelihood (PAML) (*Yang, 2007*). We discovered evidence of diversification under positive selection in these 13 species, with 68 amino acid positions changing rapidly in Xrp1 (*Figure 8A*). This result suggested that *Xrp1* was under strong positive selection for evolutionary change.

Using BLAST searches we have now identified apparent *Xrp1* orthologs in other Dipteran insects (*Figure 8B*, *Figure 8—figure supplement 1*). These homologs are the most similar genes to *Xrp1* in their respective genomes, and correspondingly *Xrp1* is their most similar *Drosophila* homolog. All share the AT-hook and bZIP domains (*Figure 8C*), and a separated Xrp1-homology domain more

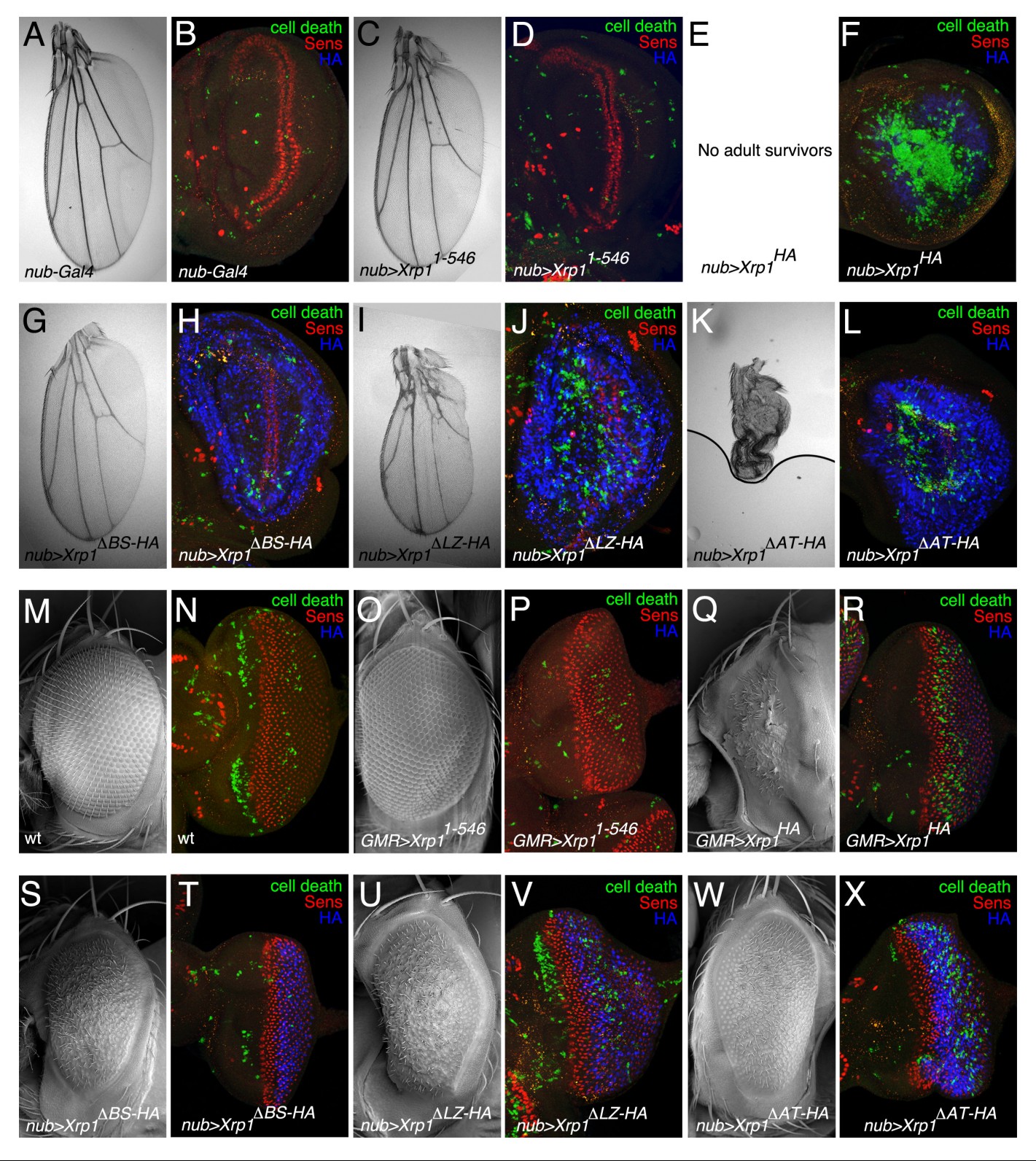

**Figure 7.** Ectopic expression of mutated Xrp1 proteins. (A–L) Over-expression of Xrp1 proteins during wing development using *nub-Gal4* at 18˚C. A,C, E,G,I,K show wings from males misexpressing the indicated proteins. Results were similar from females although the male wings were affected more. (B,D, F, H, J, L) show third instar wing imaginal discs labeled for Senseless (red) to reveal the neural differentiation pattern along the future wing margin, anti-active caspase Dcp1 (green) to reveal cell death, and anti-HA (blue) to detect expression of mutated Xrp1 proteins. The Xrp1[1-546] protein,

*Figure 7 continued on next page*

Figure 7 continued

which had little or no effect on wing development, was not detected by anti-HA because it was not tagged (D) Notably, each of the other deletion proteins was expressed more highly than the wild type (compare blue signal in panels H,J,L with F) although it can't be excluded this reflects apoptosis of many cells expressing wild type Xrp1 (F) No adults survived expression of full-length Xrp1 in wings (E). (M–X) Over-expression of Xrp1 proteins during eye development using *GMR-Gal4* at 18°C. M,O,Q,S,U,W show eyes from males misexpressing the indicated proteins. Results were similar from females. N,P, R, T, V, X show third instar eye imaginal discs labeled for Senseless (red) to reveal the retinal differentiation pattern posterior to the morphogenetic furrow, anti-active caspase Dcp1 (green) to reveal cell death, and anti-HA (blue) to detect expression of mutated Xrp1 proteins. Genotypes A,B) *nub-Gal4/+*. C,D) *nub-Gal4/+; UAS-Xrp1^{1-546}*. E,F) *nub-Gal4/+; UAS-Xrp1^{HA}*. G,H) *nub-Gal4/+; UAS-Xrp1^{ΔBR-HA}*. I,J) *nub-Gal4/+; UAS-Xrp1^{ΔLZ-HA}*. K,L) *nub-Gal4/+; UAS-Xrp1^{ΔAT-HA}*. (M,N) *w^{11-18}*. (O-P) *GMR-Gal4/+; UAS-Xrp1^{1-546}* (Q-R) *GMR-Gal4/+; UAS-Xrp1^{HA}*. (S-T) *GMR-Gal4/+; UAS-Xrp1^{ΔBR-HA}*. (U-V) *GMR-Gal4/+; UAS-Xrp1^{ΔLZ-HA}*. (W-X) *GMR-Gal4/+; UAS-Xrp1^{ΔAT-HA}*..

The online version of this article includes the following figure supplement(s) for figure 7:

**Figure supplement 1.** *Xrp1^{1-546}* expression at higher temperature.

amino-terminal in the protein, which is in fact the most highly conserved region of Xrp1 (*Figure 8D*). Outside of these domains, the *Xrp1* proteins appear to be highly divergent (*Figure 8B*, *Figure 8— figure supplement 1* ).

It is possible that rapid divergence under positive selection obscures Xrp1 homologs beyond Dipteran insects. Accordingly, we reviewed human bZIP proteins that in vitro interact with C/EBPγ, the human protein most resembling Irbp18 (*Reinke et al., 2013*; *Francis et al., 2016*). The DNA Damage Induced Transcript 3 (DDIT3) encodes a protein that shares some properties with Xrp1. Like Xrp1, DDIT3 is induced in response to DNA damage, regulates apoptosis, and heterodimerizes with C/EBPγ. DDIT3, also known as CHOP, lacks any recognizable ortholog in *Drosophila*. Interestingly, phylogenetic bootstrapping analysis has suggested that DDIT3 might be the most similar human bZip domain to that of *Drosophila* Xrp1 (*Baillon et al., 2018*). We used *GMRgal4* to express *hDDIT3* ectopically during *Drosophila* eye development. Interestingly, the ectopic expression of *hDDIT3* not only reduced the eye size like Xrp1 does (*Figure 8E*), but this phenotype was ameliorated by the simultaneous expression of RNAi against *Irbp18* (*Figure 8F*) or *Xrp1* (*Figure 8G*).

## Discussion

Heterozygous mutation of ribosomal protein genes lead to cell-autonomous, deleterious phenotypes in both flies and mammals and provide the classic example of a genotype that is eliminated from mosaics by competition (*Morata and Ripoll, 1975*; *Oliver et al., 2004*; *Clavería and Torres, 2016*). There is increasing interest in the potential roles of cell competition in mammalian development, cancer development, and in regenerative medicine (*Clavería and Torres, 2016*; *Di Gregorio et al., 2016*; *Merino et al., 2016*; *Baker, 2017*; *Maruyama and Fujita, 2017*; *Ellis et al., 2019*). A remarkable recent finding from *Drosophila* is that many of the phenotypic effects of mutating ribosomal protein genes are mediated by a putative transcription factor, Xrp1, rather than as a direct consequence of altered ribosome number (*Lee et al., 2018*). Accordingly, Xrp1 plays a key role in the elimination of *Rp* mutant cells by cell competition (*Lee et al., 2016*; *Baillon et al., 2018*; *Lee et al., 2018*). Xrp1 transcription and protein expression are elevated in *Rp* mutant cells, restricting translation, cellular growth rate, and the rate of organismal development, and enabling cell competition with nearby wild type cells (*Lee et al., 2018*). Xrp1 had previously been implicated in the DNA damage response downstream of p53 and in the transposition of P elements, and contributes to the pathology of a *Drosophila* model of Amyotrophic Lateral Sclerosis, as well as to the coordination of organ growth in flies with *Rp* gene knockdowns (*Brodsky et al., 2004*; *Akdemir et al., 2007*; *Francis et al., 2016*; *Mallik et al., 2018*; *Boulan et al., 2019*).

Xrp1 has been reported not to have homologs in other eukaryotes (*Akdemir et al., 2007*). This seems surprising given the highly conserved and fundamental roles of ribosomal proteins, and is a barrier to investigating the potential conservation of cell competition mechanisms and the roles of cell competition in mammals, for example in the development of cancer. Xrp1 binds to DNA as a heterodimer with Irbp18, the *Drosophila* homolog of the C/EBP protein family, which is a conserved protein (*Akdemir et al., 2007*; *Francis et al., 2016*) and co-purifies with it in cultured cells (*Francis et al., 2016*; *Mallik et al., 2018*). Irbp18 in turn heterodimerizes with the conserved protein dATF4, encoded by the *crc* gene in *Drosophila* (*Reinke et al., 2013*). This led us to

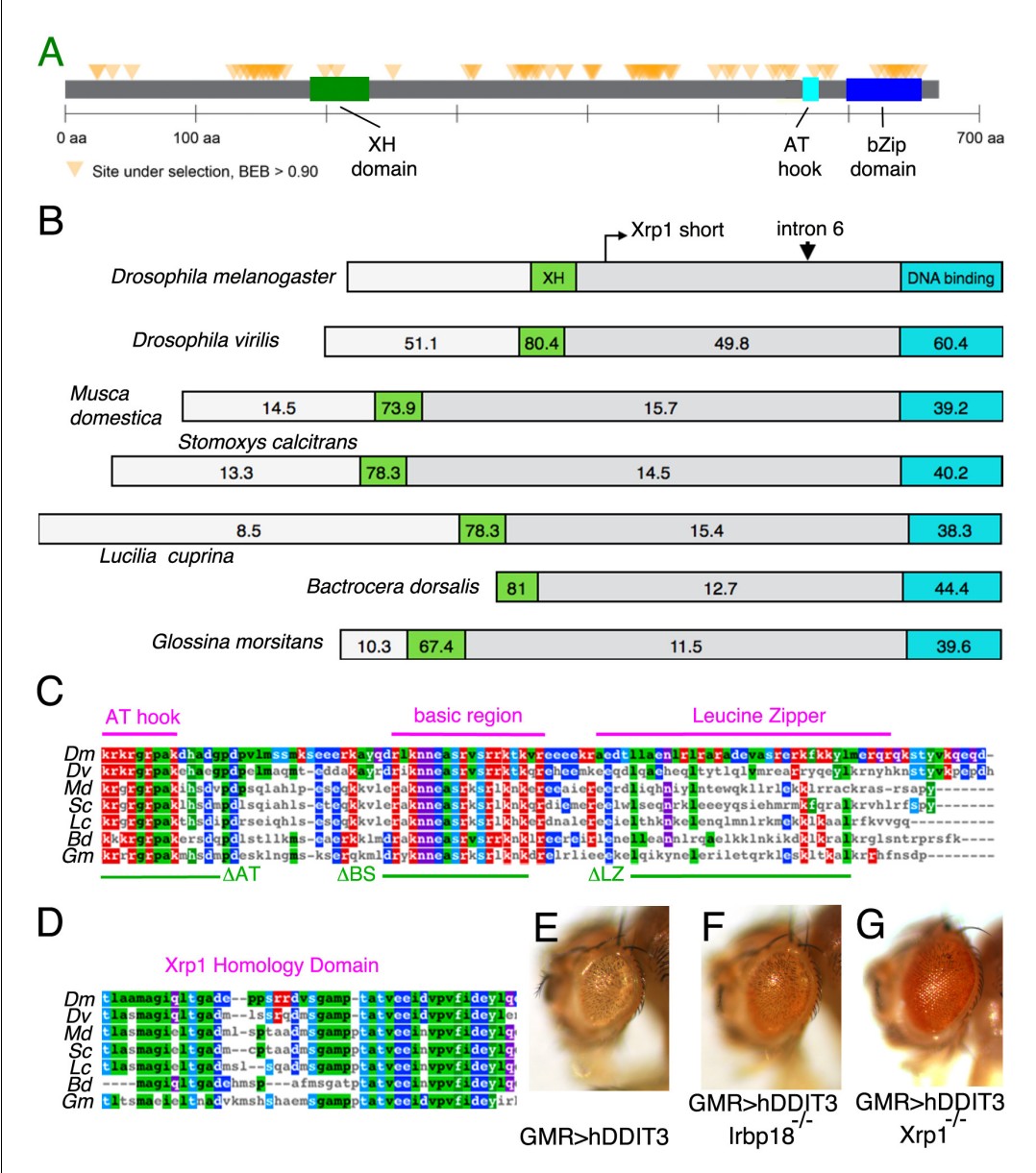

**Figure 8.** *Xrp1* gene conservation. (**A**) PAML results for 12 *Drosophila* species showing the location of sites under strong positive selection in the Xrp1 protein. (**B**) The conservation (% amino-acid identity to Xrp1 from *Drosophila melanogaster*) is plotted for *Drosophila virilis* and for predicted proteins from five other Dipterans. The C-terminal DNA-binding domain region (corresponding to amino acids 565–668 from the long form of D. melanogaster Xrp1) is highlighted in Cyan, and the more amino-terminal conserved sequence (corresponding to amino acids 189–234 from the long form of D. melanogaster) in green. The locations of the alternative amino-terminus of Xrp1 short isoforms in *D. melanogaster*, and the position of intron six whose modification was described in *Figure 6A* are indicated. (**C**) Clustal Omega alignment of the C-terminal DNA-binding domain region of insect Xrp1 sequences. The core consensus sequences defining the AT-hook domain, basic region of bZip domains, and the Leucine Zipper, are overlined in magenta. The deletions made in this study for structure-function analysis of *D. melanogaster* Xrp1 are underlined in green. (**D**) Clustal Omega alignment of the Xrp1 Homology domain in the amino-terminal portion of the insect Xrp1 sequences. The *Bactrocera dorsalis* sequence is perhaps not optimally aligned by this program. (**E**) Expression of hDDIT3 under GMR-Gal4 control (at 25°C) reduces eye size (compare the effects of Xrp1 overexpression, *Figure 6A,C*). (**F**) Eye size reduction by hDDIT3 depended in part on the *Drosophila irbp18* gene. (**G**) Eye size reduction by hDDIT3 depended in part on the *Drosophila Xrp1* gene.

The online version of this article includes the following figure supplement(s) for figure 8:

**Figure supplement 1.** Clustal O alignment of protein sequences from the Dipteran insects *Drosophila melanogaster*, *Drosophila virilis*, *Musca domestica*, *Stomoxys calcitrans*, *Lucilia cuprina*, *Bactrocera dorsalis*, and *Glossina morsitans*.

investigate whether it is the Xrp1 heterodimer with the conserved Irbp18 protein that functions in $Rp^{+/-}$ cells, and if so whether Xrp1/Irbp18 acts positively; alternatively, Xrp1 could act as a competitive inhibitor of Irbp18 function with its other partner, dAtf4/Crc, in which case Xrp1 could represent a *Drosophila*-specific regulator of a more conserved pathway.

Our data provide overwhelming genetic evidence that Xrp1 does function along with Irbp18. Like *Xrp1* mutations, *irbp18* mutation suppressed multiple effects of *Rp* mutations, including the elimination of $Rp^{+/-}$ mutant cells by competitive apoptosis in the proximity of wild type cells and the reduced growth of $Rp^{+/-}$ wing cells. Like *Xrp1*, *irbp18* was also required for the prompt disappearance of $Rp^{-/-}$ cell clones, which survived in the *irbp18* mutant background. All these data were consistent with the model that Xrp1/Irbp18 heterodimers are the active species in *Rp* mutant cells and were inconsistent with the idea that Xrp1 might act as a competitive inhibitor of other Irbp18-containing species, as this would have predicted that Irb18 mutations would have had phenotypes opposite to those of Xrp1.

The phenotype of Crc knockdown is different from those of *Xrp1* and *irbp18* mutations. Whereas *Xrp1* and *irbp18* mutations enhance the growth and competitiveness of $Rp^{+/-}$ cells, *crc* knockdown greatly diminished growth and survival of $Rp^{+/-}$ cells. If Xrp1 was a *Drosophila*-specific competitive inhibitor of a conserved Crc/Irbp18 heterodimer that was required for growth, both *irbp18* and *crc* mutants would show reduced growth, similar to $Rp^{+/-}$ genotypes. In contrast to this, *irbp18* mutants have little phenotype except in $Rp^{+/-}$ genotypes, where their effects closely resemble those of *Xrp1* mutants. Also, whereas *crc* knockdown strongly and cell-autonomously affected the growth of $Rp^{+/-}$ cells, it had less effect on $Rp^{+/+}$ cells.

In addition to these findings in loss-of-function experiments, we also found that Xrp1 over-expression phenotypes depended on IRBP18, as would be expected if these proteins function together. We also found that Xrp1 over-expression at higher temperatures resulted in a still stronger eye phenotype where simultaneous *irbp18* mutation restored normal eye size but did not completely restore eye morphology. This is consistent with some IRBP18-independent component to ectopic Xrp1 function that is either cold-sensitive or only apparent at the highest ectopic expression levels. There is not yet any evidence whether these over-expression effects are physiologically relevant.

Taken together, our findings suggest that the Xrp1/IRBP18 and Crc/IRBP18 heterodimers have independent and perhaps unrelated functions. Consistent with this, ectopic expression of IRBP18 had no phenotypic effect, suggesting that IRBP18 is normally made in excess, so that it is Xrp1 that is limiting for the growth inhibiting activities of the Xrp1/IRBP18 heterodimers, which do not impact IRBP18 availability sufficiently to affect Crc/Irbp18 functions.

As expected if Xrp1 functions in a heterodimer, the basic and Leucine Zipper domains were important for Xrp1 function, as was the AT hook. In over-expression assays only, there could be reduced activity of proteins deleted for any of these domains individually, but not of a truncation that precedes them all. When encoded from the endogenous locus, basic and AT-hook domains appeared absolutely required.

Mutual auto-regulation may be a significant feature of Xrp1 and IRBP18 function. As noted previously, the elevated *Xrp1* and *irbp18* transcription observed in $Rp^{+/-}$ wing discs is dependent on *Xrp1* function (*Baillon et al., 2018*; *Lee et al., 2018*)(*Figure 3Q–R*). Here we show that IRBP18 protein levels are also elevated in $Rp^{+/-}$ cells in an Xrp1-dependent fashion, and that *irbp18* is also required for the autoregulation. In principle, autoregulation could have been the major or indeed the only transcriptional function of the Xrp1/IRBP18 heterodimer, ie perhaps these proteins could control cell competition through other mechanisms once levels were sufficient. This cannot be completely correct, however, because Xrp1 was still substantially dependent on the *irbp18* gene and on the Leucine Zipper and DNA binding domains when expressed using GAL4-driven transgenes that are independent of auto-regulation, so by-passing the requirement of auto-regulation does not relieve the requirements for heterodimerization and DNA binding domains. It is also worth noting that *Xrp1* and *irbp18* are both required to promptly eliminate $Rp^{-/-}$ cells, where their expression does not require auto-regulation. Although implicating other transcriptional targets of Xrp1/IRBP18 in $Rp^{+/-}$ and $Rp^{-/-}$ cells, these studies do not rule out other functions besides transcription.

Previously it was thought that Xrp1 was restricted to the genus *Drosophila*, a surprising finding for a protein that has an important cellular function. We found, however, that *Xrp1* genes have been under strong positive selection for rapid evolutionary change. Recurrent positive selection is often the sign of an evolutionary arms race, such as are often driven by host-pathogen interactions, sexual

competition, or intra-genomic conflict (*Valen, 1973*; *Sawyer et al. (2005)*; *Clark et al. (2006)*; *Elde and Malik (2009)*; *Levine and Malik (2013)*; *Enard et al., 2016*). Possibly pathogens target Xrp1 to promote growth and survival of infected cells. It is interesting that Xrp1 is already documented to interact with one transposable element, the P element (*Francis et al., 2016*). However, none of these scenarios for positive selection, or indeed additional possibilities, can yet be ruled out.

Rapid divergence makes homology difficult to detect, and accordingly we now identify divergent *Xrp1* homologs in other Dipteran insects that have not previously been annotated because their sequence similarities are restricted to the key DNA-binding portion of the protein, and to a more amino-terminal Xrp1-homology domain. The failure to identify still more distant homologs might be genuine, or might reflect further divergence beyond our ability to recognize homology. Mammals do contain other members of the C/EBP protein family without identified *Drosophila* homologs, and we show that DDIT3 (aka CHOP and C/EBP-Z) can generate a similar phenotype to Xrp1 when expressed in *Drosophila*. Interestingly C/EBP-α, one of the mammalian proteins more related to Irbp18, has been implicated in a cell competition-like phenomenon, the elimination of cells from the multipotent hematopoietic stem cell niche following irradiation (*Fleenor et al., 2015*).

## Materials and methods

### Fly strains and clonal analysis

Flies were reared on standard medium at 25°C. The genetic strains used are described in the Key Resources Table (*Supplementary file 1*).

Mitotic recombination was induced 24–72 hr after egg laying by a 1 hr heat shock at 37°C and the larvae dissected 72 or 96 hr later. Genotypes of the analyzed larvae were as follows:

- *wt* MARCM clones overexpressing *Xrp1*: *y w hs-FLP; tubP-GAL4 UAS-mCD8::GFP/UAS-Xrp1; FRT82B arm-LacZ/FRT82B tubP-GAL80*.
- *wt* MARCM clones overexpressing *CG6272*: *y w hs-FLP; tubP-GAL4 UAS-mCD8::GFP/UAS-CG6272; arm-LacZ FRT80/tubP-GAL80 FRT80*.
- *CG6272*$^{f05006}$ MARCM clones: *y w hs-FLP; tubP-GAL4 UAS-mCD8::GFP/+*; CG6272*$^{f05006}$ arm-LacZ FRT80/tubP-GAL80 FRT80*. (*UAS-Xrp1* instead of *+** for the ectopic expression of *Xrp1* in *CG6272*$^{f05006}$ clones).
- *Xrp1*$^{M2-73}$ MARCM clones in *M/+* background: *y w hs-FLP; tubP-GAL4 UAS-mCD8::GFP/FRT42 ubi-GFP M(2)56F; FRT82B Xrp1*$^{M2-73}$/FRT82B tubP-GAL80*.
- *CG6272*$^{f05006}$ MARCM clones in *M/+* background: *y w hs-FLP; tubP-GAL4 UAS-mCD8::GFP/ FRT42 ubi-GFP M(2)56F; CG6272*$^{f05006}$ FRT80/tubP-GAL80 FRT80*.
- *M/+* clones in *wt* background: *w M(1)Bld/hs FLP; P[RpL36+ w+] arm-LacZ FRT80/FRT80** (*CG6272*$^{f05006}$ instead of * for the generation of *M/+* clones homozygous for *CG6272*$^{f05006}$).
- *wt* clones in *M/+* background: *y w hs-FLP; FRT42 ubi-GFP M(2)56F/FRT42; +*/+* (instead of +* the following alleles were used to study their effect on cell competition: *CG6272*$^{f05006}$, *Xrp1*$^{HA}$, *Xrp1*$^{ΔΔT-HA}$ and *Xrp1*$^{ΔBS-HA}$).
- *CG6272*$^{f05006}$ clones in *wt* and *M/+* backgrounds: *y w hs-FLP; FRT42 ubi-GFP M(2)56F*/+; CG6272*$^{f05006}$ FRT80/arm LacZ FRT80* (+ instead of * for the *wt* background).

### Replicates and power analysis

We did not perform a prior power analysis for the quantitative comparisons shown in *Figure 1B,C*, *Figure 1G,H*, and *Figure 2A–D* because previous studies of comparable *Xrp1* mutant genotypes had established that statistically-significant results could be obtained from ~5 preparations (*Lee et al., 2018*). For other experiments where statistical analysis was not reported at least two biological replicates were generally performed, ie the complete immunostaining and imaging was performed twice independently. For the over-expression experiments shown in *Figure 7*, dozens of adult flies were examined for each genotype and two technical replicates for each recorded by confocal microscopy after immunohistochemistry.

## Immunohistochemistry

Antibody staining on discs and most antibodies were described previously (*Baker et al., 2014*). Antibodies and their dilutions are described in the Key Resources Table (*Supplementary file 1*). Fluorescent images were captured with Leica TCS SP5 AND SP8 scanning confocal microscopes and processed using Image J64 and Adobe Photoshop CS5.

## Phylogenetic analysis

To analyze the evolutionary history of Xrp1, we compared sequences from 13 species of *Drosophila* ranging from *D. melanogaster* to *D. annanase*. We chose these species based on strong homology hits in tBLASTn to the *D. melanogaster* Xrp1 protein coding sequence from the publicly available *Drosophila* genomes. To rest for recurrent positive selection across these species of flies, we used Phylogenetic Analysis by Maximum Likelihood (PAML) v4.9 [1]. We used three NSsites models in our analysis – Model 7, Model 8, and Model 8a – to discriminate between purifying, positive, and neutral selection respectively. We used a log-likelihood ratio test to determine if there was a significant difference in the fit of the models to our data.

## DNA constructs and transgenic flies

The *Xrp1^HA^*, *Xrp1^ΔAT-HA^* and *Xrp1^ΔBS-HA^* alleles were obtained from the strain *Xrp1^attPFlox^* following the method described (*Baena-Lopez et al., 2013*).

## Generation of the *Xrp1^attPflox^* allele

5' and 3' homology arms (HA) were amplified by PCR using as a template genomic DNA from *D. melanogaster yw* flies and the following primer combinations

 1) 5'HA, Length 1088
 F-<u>GAATTC</u>TGCACAATACAGGGTGTTCT (*EcoR*I)
 R-<u>CATATG</u>AAAGAGCTGACAATTTTTCGGTAC (*Nde*I)
 2) 3'HA, Length 1174
 F-<u>ACTAGT</u>TGAAGTCGAAATATCCAGCACAGCC (*Spe*I)
 R-<u>CCTAGG</u>AAGAACGTTATAGAACCCATTCAAA (*Avr*II)

 The 5'HA and 3'HA DNA fragments were cloned, respectively, into the 5' and 3' MCS (Multiple Cloning Site) of the pTV^Cherry^ vector and the resulting construct (pTV-Xrp1) was used for germline transformation of yw flies (Rainbow transgenics). Transgenic male flies harbouring the pTV-Xrp1 construct were crossed to females containing the *Flippase* and *I-SceI* genes under the control of the *heatshock* promoter (Bloomington stock #25679). Larvae from this cross were heat-shocked to induce the excision and linearization of the *5'HA-attP-mini White-3'HA* cassette and adults with mottled red eyes were crossed to *ubi-gal4* flies to eliminate false positives (see *Baena-Lopez et al., 2013*). Red eye flies from this cross were considered candidates and further analysed by PCR and Sanger sequencing to confirm the reintegration of the cassette into the *Xrp1* locus by homologous recombination.

 The *Xrp1^attPflox^* allele behaves during cell competition as a LOF allele (Suppl. Fig. 1).

## Generation of *Xrp1^attPloxP^*

The *5'HA-attP-mini White-3'HA* cassette integrated onto the *Xrp1* locus contains *LoxP* sites flanking the *mini-White* gene. By crossing red eyes *Xrp1^attPflox^* flies with *hs-Cre* flies (Bloomington #851), white eyes *Xrp1^attPloxP^* flies lacking the mini-White gene were obtained. The *Xrp1^attPloxP^* allele behaves as a *wt* allele during cell competition (Suppl. Fig. 1).

| | |
|---|---|
| *Xrp1* | **CGG**GTAAGTATCTGTAAAGAACAGATTAAATGGTGTACCGAAAAATTGTCAGCTCTTT-- |
| *Xrp1^attPloxP^* | **CGG**GTAAGTATCTGTAAAGAACAGATTAAATGGTGTACCGAAAAATTGTCAGCTCTTT**CA** |
| | ************************************************************ |
| *Xrp1* | ------------------------------------------------------------ |
| *Xrp1^attPloxP^* | **TATG**CAGGTACC**GTAGTGCCCCAACTGGGGTAACCTTTGAGTTCTCTCAGTTGGGGGCGT** |

*Continued on next page*

| | |
|---|---|
| *Xrp1* | ------------------------------------------------TGAAGTCGAAA |
| *Xrp1^attPloxP^* | **AG**TGTACC**ATAACTTCGTATAATGTATGCTATACGAAGTTAT**C**ACTAGT**TGAAGTCGAAA |
| | \*\*\*\*\*\*\*\*\*\* |
| *Xrp1* | TATCCAGCACAGCCTAATTTACTTAATCTCTCCTATTTGCAG**TTC** |
| *Xrp1^attPloxP^* | TATCCAGCACAGCCTAATTTACTTAATCTCTCCTATTTGCAG**TTC** |
| | \*\*\*\*\*\*\*\*\*\*\*\*\*\*\*\*\*\*\*\*\*\*\*\*\*\*\*\*\*\*\*\*\*\*\*\*\*\*\*\*\*\*\*\*\*\*\*\* |

The end of exon 6 and beginning of exon 7 are shown in black bold letters.
Introduced restriction sites: *Nde*I: **CATATG** and *Spe*I: **ACTAGT**
Significant sequences: attP and LoxP sites.

## Generation of the *Xrp1^HA^*, *Xrp1^ΔAT-HA^*, *Xrp1^ΔLZ-HA^* and *Xrp1^ΔBS-HA^*

The *attP* site present in the *Xrp1^attPloxP^* allele was used as a platform to introduce modifications in the last exon of the *Xrp1* gene using PhiC31-mediated recombination. For this purpose, flies having a source of the PhiC31 integrase gene on the X chromosome under the control of a germline-specific promoter (Bloomington #40161), were crossed to *Xrp1^attPloxP^* flies to generate the recipient strain *y¹ M(vas-int.Dm)ZH2A; Xrp1^attPloxP^*. This strain was used in germline transformation experiments (Bestgene) using pRIVW-based plasmids that contain the *attB* site upstream of the modified last exon of *Xrp1*. The pRIVW vector derives from the reintegration vector RIV^FRT MCS FRT MCS3^ (*Baena-Lopez et al., 2013*) by cloning the *mini-White* gene between the *Pst*I and *Hpa*I restriction sites. The modifications introduced in the last exon of the *Xrp1* gene were generated by PCR using proper primer combinations. Each modified version of the Xrp1 last exon was generated using 2 PCR products (5′ and 3′ products) that share a common restriction site (<u>S</u>hared <u>R</u>estriction <u>S</u>ite) at the place where the modification was introduced. The 5′ products (*Spe*I-SRS) were cloned into the pRIVW vector digested with *Xba*I (partial digestion) and *SRS*. The 3′ products (SRS-*Avr*II) were subsequently cloned using the *SRS* and *Avr*II restriction sites. By cloning the PCR products between *Xba*I and *Avr*II restriction sites, 2 FRT sites present in the pRIVW vector are eliminated. The final constructs were injected into *y¹ M(vas-int.Dm)ZH2A; Xrp1^attPloxP^* flies and the candidate red eye transformants analysed by PCR.

1) Introduction of a HA-tag at the C-terminal end of *Xrp1*
-5′PCR product
F-<u>ACTAGT</u>TGAAGTCGAAATATCCAGCACAGCC (*Spe*I)
R-C<u>GAATTC</u>TCAGTC**CGCGTAGTCGGGGACGTCGTAGGGGTA**CTGCTCCTGCTTAACG
TAAGTGCTC (*EcoR*I)
-3′PCR
F- G<u>GAATTC</u>GATTCTCGTAGCCATAGAATAGAAATTGCGG (*EcoR*I)
R- <u>CCTAGG</u>AAGAACGTTATAGAACCCATTCAAA (*Avr*II)

Relevant sequences of the resulting pRIVW-Xrp1-HA vector: **attB site** in **bold letters**, HA sequence, _Xrp1 Exon7_ is underlined.

```
ATAATTTGCGAGTACGCAAAGCTATGCATGTAATGAATTGCCGCGGTGCGGGTGCCAGGGCGT
GCCCTTGGGCTCCCCGGGCGCGTACTCCCGTACGACGAAGTTCCTATTCCGAAGTTCCTATTC
TCTAGTTGAAGTCGAAATATCCAGCACAGCCTAATTTACTTAATCTCTCCTATTTGCAGTTCG
AGGCTCCCACCACAAGTACCAGTACCAGCACCAGCAACACTTCCATCAGCAGCAGCACCCATG
CAGACATCGTAGAGGATCTACGCAGTGCCGAAGAGGAGACTACCACCGACTTCTCGGCGCCGA
ACACACCACACAGCAACTACTCAGCCAGCTCCAGCTGTGCGGCGCCCACCTGCCAGACCGGCT
ATGGTGGCTTCCTCACTGCTCCCACCTCGCCCGCCTATTCAACAGCCAGTACGTCCGTGTTCA
GCCCTTCGCCAGCCAGCGGAATCAGCGGCAAGCGGAAGCGCGGTCGCCCTGCCAAGGATCATG
CCGACGGTCCCGATCCCGTGCTCATGTCAAGCATGAAAAGCGAGGAGGAGAGGAAGGCCTACC
AGGACAGACTCAAGAACAACGAAGCGAGCCGCGTATCGCGCCGGAAGACGAAGGTGCGCGAAG
AGGAGGAGAAGCGCGCCGAGGACACGCTATTGGCCGAGAATCTGCGACTGCGTGCCCGGGCCG
ACGAAGTGGCCTCCCGGGAACGGAAGTTCAAGAAGTACCTGATGGAGCGCCAGCGGCAGAAGA
GCACTTACGTTAAGCAGGAGCAGTACCCCTACGACGTCCCCGACTACGCGGACTGAGAATTCG
ATTCTCGTAGCCATAGAATAGAAATTGCGGAGCATAAGCCTGTCATAGTACCTATCGTCGTCA
TCCAATTGTTTAGTTAGTTAAGTTGTCGTTGAAGAAGTCGAGAAGCAGAAATCAAATTTCCAT
TTCCATGTGGACATCAGTTAAGCCACTTATTAGGCATAGTGATGGCGCCCATAGTGCGCGAGG
AGTGTTTAACTTTAATTTTACTTTTATTATGTTTGAGAAAACTAAAAAATTTTTATACCCACG
GATATGTTTTTCGTCCAAAGCGTAATCAAAAAAAAAAAAAGCACAACAAAAAAATGATTACGA
GTTCGTATGTTTTAACCCCATTGATGAACAAAAAAAAGTTTAAAAAATGTCAACAAAAAAATA
TGCTATTGTTAAGAACAATTGAAATGTATATTTTACAGATATCAACTTATATAACCGTAATTG
TTTGAATAAAAAGCATGTTCAAATAAACGTGACAAAATCAACTAACTTTGTATATTTGAATGG
GTTCTATAACGTTCTT
```

The STOP codon is shown in bold black underlined letters. The DNA binding domains and the dimerization domain are shown in bold letters: AT Hook, _Basic domain_ (_italicized_) and Leucine zipper. Introduced restriction site _EcoR_I is italicized: _GAATTC_

The 5' cloning site of the 5'PCR product is a _Xba_I/_Spe_I fusion site, shown italicized: _TCTAGT_.

The pRIVW-Xrp1-HA vector was used as a template to generate the deletions using the following primer combinations:

2) Deletion in the AT-Hook domain

-5'PCR product

F- ACTAGTTGAAGTCGAAATATCCAGCACAGCC (_Spe_I)

R- AGATCTGCCGCTGATTCCGCTGGCTG (_SRS: Bgl_II)

-3'PCR product

F- AGATCTCCCGATCCCGTGCTCATGTC (_SRS: Bgl_II)

R- CCTAGGAAGAACGTTATAGAACCCATTCAAA (_Avr_II)

Exon7 alignment

| | |
|---|---|
| _Xrp1_ | TTCGAGGCTCCCACCACAAGTACCAGTACCAGCACCAGCAACACTTCCATCAGCAGCAGC |
| _Xrp1_ΔAT-HA | TTCGAGGCTCCCACCACAAGTACCAGTACCAGCACCAGCAACACTTCCATCAGCAGCAGC |
| | ************************************************************ |
| _Xrp1_ | ACCCATGCAGACATCGTAGAGGATCTACGCAGTGCCGAAGAGGAGACTACCACCGACTTC |
| _Xrp1_ΔAT-HA | ACCCATGCAGACATCGTAGAGGATCTACGCAGTGCCGAAGAGGAGACTACCACCGACTTC |
| | ************************************************************ |
| _Xrp1_ | TCGGCGCCGAACACACCACACAGCAACTACTCAGCCAGCTCCAGCTGTGCGGCGCCCACC |
| _Xrp1_ΔAT-HA | TCGGCGCCGAACACACCACACAGCAACTACTCAGCCAGCTCCAGCTGTGCGGCGCCCACC |
| | ************************************************************ |
| _Xrp1_ | TGCCAGACCGGCTATGGTGGCTTCCTCACTGCTCCCACCTCGCCCGCCTATTCAACAGCC |
| _Xrp1_ΔAT-HA | TGCCAGACCGGCTATGGTGGCTTCCTCACTGCTCCCACCTCGCCCGCCTATTCAACAGCC |
| | ************************************************************ |
| _Xrp1_ | AGTACGTCCGTGTTCAGCCCTTCGCCAGCCAGCGGAATCAGCGGCAAGCGGAAGCGCGGT |
| _Xrp1_ΔAT-HA | AGTACGTCCGTGTTCAGCCCTTCGCCAGCCAGCGGAATCAGCGGC-------------- |
| | ********************************************* |
| _Xrp1_ | CGCCCTGCCAAGGATCATGCCGACGGTCCCGATCCCGTGCTCATGTCAAGCATGAAAAGC |

_Continued on next page_

| | |
|---|---|
| *Xrp1^ΔAT-HA* | -**AGATCT**-------------------CCCGATCCCGTGCTCATGTCAAGCATGAAAAGC |
| | ****************************** |
| *Xrp1* | GAGGAGGAGAGGAAGGCCTACCAG***GACAGACTCAAGAACAACGAAGCGAGCCGCGTATCG*** |
| *Xrp1^ΔAT-HA* | GAGGAGGAGAGGAAGGCCTACCAGGACAGACTCAAGAACAACGAAGCGAGCCGCGTATCG |
| | ************************************************************ |
| *Xrp1* | ***CGCCGGAAGACGAAGGTGCGCGAAGAGGAGGAGAAGCGC***GCCGAGGACACGCTATTGGCC |
| *Xrp1^ΔAT-HA* | CGCCGGAAGACGAAGGTGCGCGAAGAGGAGGAGAAGCGCGCCGAGGACACGCTATTGGCC |
| | ************************************************************ |
| *Xrp1* | GAGAATCTGCGACTGCGTGCCCGGGCCGACGAAGTGGCCTCCCGGGAACGGAAGTTCAAG |
| *Xrp1^ΔAT-HA* | GAGAATCTGCGACTGCGTGCCCGGGCCGACGAAGTGGCCTCCCGGGAACGGAAGTTCAAG |
| | ************************************************************ |
| *Xrp1* | AAGTACCTGATGGAGCGCCAGCGGCAGAAGAGCACTTACGTTAAGCAGGAGCAG------ |
| *Xrp1^ΔAT-HA* | AAGTACCTGATGGAGCGCCAGCGGCAGAAGAGCACTTACGTTAAGCAGGAGCAG**TACCCC** |
| | ****************************************************** |
| *Xrp1* | -------------------GAC**TGA**------GATTCTCGTAGCCATAGAATAGAAATT |
| *Xrp1^ΔAT-HA* | **TACGACGTCCCCGACTACGCG**GAC**TGA**GAATTCGATTCTCGTAGCCATAGAATAGAAATT |
| | ******        ************************** |
| *Xrp1* | GCGGAGCATAAGCCTGTCATAGTACCTATCGTCGTCATCCAATTGTTTAGTTAGTTAAGT |
| *Xrp1^ΔAT-HA* | GCGGAGCATAAGCCTGTCATAGTACCTATCGTCGTCATCCAATTGTTTAGTTAGTTAAGT |
| | ************************************************************ |
| *Xrp1* | TGTCGTTGAAGAAGTCGAGAAGCAGAAATCAAATTTCCATTTCCATGTGGACATCAGTTA |
| *Xrp1^ΔAT-HA* | TGTCGTTGAAGAAGTCGAGAAGCAGAAATCAAATTTCCATTTCCATGTGGACATCAGTTA |
| | ************************************************************ |
| *Xrp1* | AGCCACTTATTAGGCATAGTGATGGCGCCCATAGTGCGCGAGGAGTGTTTAACTTTAATT |
| *Xrp1^ΔAT-HA* | AGCCACTTATTAGGCATAGTGATGGCGCCCATAGTGCGCGAGGAGTGTTTAACTTTAATT |
| | ************************************************************ |
| *Xrp1* | TTACTTTTATTATGTTTGAGAAAACTAAAAAATTTTTATACCCACGGATATGTTTTTCGT |
| *Xrp1^ΔAT-HA* | TTACTTTTATTATGTTTGAGAAAACTAAAAAATTTTTATACCCACGGATATGTTTTTCGT |
| | ************************************************************ |
| *Xrp1* | CCAAAGCGTAATCAAAAAAAAAAAAAGCACAACAAAAAAATGATTACGAGTTCGTATGTT |
| *Xrp1^ΔAT-HA* | CCAAAGCGTAATCAAAAAAAAAAAAAGCACAACAAAAAAATGATTACGAGTTCGTATGTT |
| | ************************************************************ |
| *Xrp1* | TTAACCCCATTGATGAACAAAAAAAAGTTTAAAAAATGTCAACAAAAAAATATGCTATTG |
| *Xrp1^ΔAT-HA* | TTAACCCCATTGATGAACAAAAAAAAGTTTAAAAAATGTCAACAAAAAAATATGCTATTG |
| | ************************************************************ |
| *Xrp1* | TTAAGAACAATTGAAATGTATATTTTACAGATATCAACTTATATAACCGTAATTGTTTGA |
| *Xrp1^ΔAT-HA* | TTAAGAACAATTGAAATGTATATTTTACAGATATCAACTTATATAACCGTAATTGTTTGA |
| | ************************************************************ |
| *Xrp1* | ATAAAAAGCATGTTTCAATAAACGTGACAAAATCAACT |
| *Xrp1^ΔAT-HA* | ATAAAAAGCATGTTTCAATAAACGTGACAAAATCAACT |
| | ************************************** |

The STOP codon is shown in bold black letters. Introduced restriction sites *Bgl*II: AGATCT and *Eco*RI: GAATTC. Significant sequences: AT-hook domain, **basic domain** (italicized **in bold**) and HA tag sequence.
3) Deletion in the Basic domain
-5'PCR product
F- <u>ACTAGT</u>TGAAGTCGAAATATCCAGCACAGCC (*Spe*I)

R- <u>GCTAGC</u>CTGGTAGGCCTTCCTCTCCTC (*SRS: NheI*)
-3'PCR product
F- <u>GCTAGC</u>GTGCGCGAAGAGGAGGAGAAG (*SRS: NheI*)
R- <u>CCTAGG</u>AAGAACGTTATAGAACCCATTCAAA (*AvrII*)
Exon7 alignment

| | |
|---|---|
| *Xrp1* | TTCGAGGCTCCCACCACAAGTACCAGTACCAGCACCAGCAACACTTCCATCAGCAGCAGC |
| *Xrp1ΔBS-HA* | TTCGAGGCTCCCACCACAAGTACCAGTACCAGCACCAGCAACACTTCCATCAGCAGCAGC |
| | ************************************************************ |
| *Xrp1* | ACCCATGCAGACATCGTAGAGGATCTACGCAGTGCCGAAGAGGAGACTACCACCGACTTC |
| *Xrp1ΔBS-HA* | ACCCATGCAGACATCGTAGAGGATCTACGCAGTGCCGAAGAGGAGACTACCACCGACTTC |
| | ************************************************************ |
| *Xrp1* | TCGGCGCCGAACACACCACACAGCAACTACTCAGCCAGCTCCAGCTGTGCGGCGCCCACC |
| *Xrp1ΔBS-HA* | TCGGCGCCGAACACACCACACAGCAACTACTCAGCCAGCTCCAGCTGTGCGGCGCCCACC |
| | ************************************************************ |
| *Xrp1* | TGCCAGACCGGCTATGGTGGCTTCCTCACTGCTCCCACCTCGCCCGCCTATTCAACAGCC |
| *Xrp1ΔBS-HA* | TGCCAGACCGGCTATGGTGGCTTCCTCACTGCTCCCACCTCGCCCGCCTATTCAACAGCC |
| | ************************************************************ |
| *Xrp1* | AGTACGTCCGTGTTCAGCCCTTCGCCAGCCAGCGGA**ATCAGCGGCAAGCGGAAGCGCGGT** |
| *Xrp1ΔBS-HA* | AGTACGTCCGTGTTCAGCCCTTCGCCAGCCAGCGGAATCAGCGGCAAGCGGAAGCGCGGT |
| | ************************************************************ |
| *Xrp1* | **CGCCCTGCCAAGGATCAT**GCCGACGGTCCCGATCCCGTGCTCATGTCAAGCATGAAAAGC |
| *Xrp1ΔBS-HA* | CGCCCTGCCAAGGATCATGCCGACGGTCCCGATCCCGTGCTCATGTCAAGCATGAAAAGC |
| | ************************************************************ |
| *Xrp1* | GAGGAGGAGAGGAAGGCCTACCAG**GACAGACTCAAGAACAACGAAGCGAGCCGCGTATCG** |
| *Xrp1ΔBS-HA* | GAGGAGGAGAGGAAGGCCTACCAG-------------------***GCTAGC***---------- |
| | *********************** |
| *Xrp1* | **CGCCGGAAGACGAAGGTGCGCGAAGAGGAGGAGAAGCGC**GCCGAGGACACGCTATTGGCC |
| *Xrp1ΔBS-HA* | --------------GTGCGCGAAGAGGAGGAGAAGCGCGCCGAGGACACGCTATTGGCC |
| | ******************************************** |
| *Xrp1* | GAGAATCTGCGACTGCGTGCCCGGGCCGACGAAGTGGCCTCCCGGGAACGGAAGTTCAAG |
| *Xrp1ΔBS-HA* | GAGAATCTGCGACTGCGTGCCCGGGCCGACGAAGTGGCCTCCCGGGAACGGAAGTTCAAG |
| | ************************************************************ |
| *Xrp1* | AAGTACCTGATGGAGCGCCAGCGGCAGAAGAGCACTTACGTTAAGCAGGAGCAG------ |
| *Xrp1ΔBS-HA* | AAGTACCTGATGGAGCGCCAGCGGCAGAAGAGCACTTACGTTAAGCAGGAGCAG**TACCCC** |
| | ***************************************************** |
| *Xrp1* | -------------------GAC**TGA**------GATTCTCGTAGCCATAGAATAGAAATT |
| *Xrp1ΔBS-HA* | **TACGACGTCCCCGACTACGCG**GAC**TGA*GAATTC***GATTCTCGTAGCCATAGAATAGAAATT |
| | ****** ************************* |
| *Xrp1* | GCGGAGCATAAGCCTGTCATAGTACCTATCGTCGTCATCCAATTGTTTAGTTAGTTAAGT |
| *Xrp1ΔBS-HA* | GCGGAGCATAAGCCTGTCATAGTACCTATCGTCGTCATCCAATTGTTTAGTTAGTTAAGT |
| | ************************************************************ |
| *Xrp1* | TGTCGTTGAAGAAGTCGAGAAGCAGAAATCAAATTTCCATTTCCATGTGGACATCAGTTA |
| *Xrp1ΔBS-HA* | TGTCGTTGAAGAAGTCGAGAAGCAGAAATCAAATTTCCATTTCCATGTGGACATCAGTTA |
| | ************************************************************ |
| *Xrp1* | AGCCACTTATTAGGCATAGTGATGGCGCCCATAGTGCGCGAGGAGTGTTTAACTTTAATT |
| *Xrp1ΔBS-HA* | AGCCACTTATTAGGCATAGTGATGGCGCCCATAGTGCGCGAGGAGTGTTTAACTTTAATT |

*Continued on next page*

|  | `************************************************************` |
|---|---|
| *Xrp1* | TTACTTTTATTATGTTTGAGAAAACTAAAAAATTTTTATACCCACGGATATGTTTTTCGT |
| *Xrp1*<sup>ΔBS-HA</sup> | TTACTTTTATTATGTTTGAGAAAACTAAAAAATTTTTATACCCACGGATATGTTTTTCGT |
|  | `************************************************************` |
| *Xrp1* | CCAAAGCGTAATCAAAAAAAAAAAAAGCACAACAAAAAAATGATTACGAGTTCGTATGTT |
| *Xrp1*<sup>ΔBS-HA</sup> | CCAAAGCGTAATCAAAAAAAAAAAAAGCACAACAAAAAAATGATTACGAGTTCGTATGTT |
|  | `************************************************************` |
| *Xrp1* | TTAACCCCATTGATGAACAAAAAAAAGTTTAAAAAATGTCAACAAAAAAATATGCTATTG |
| *Xrp1*<sup>ΔBS-HA</sup> | TTAACCCCATTGATGAACAAAAAAAAGTTTAAAAAATGTCAACAAAAAAATATGCTATTG |
|  | `************************************************************` |
| *Xrp1* | TTAAGAACAATTGAAATGTATATTTTACAGATATCAACTTATATAACCGTAATTGTTTGA |
| *Xrp1*<sup>ΔBS-HA</sup> | TTAAGAACAATTGAAATGTATATTTTACAGATATCAACTTATATAACCGTAATTGTTTGA |
|  | `************************************************************` |
| *Xrp1* | ATAAAAAGCATGTTTCAATAAACGTGACAAAATCAACT |
| *Xrp1*<sup>ΔBS-HA</sup> | ATAAAAAGCATGTTTCAATAAACGTGACAAAATCAACT |
|  | `**************************************` |

The STOP codon is shown in bold black letters. Introduced restriction sites italicized in bold *Nhe*I: **GCTAGC** and *EcoR*I: **GAATTC**. Significant sequences: AT-hook domain, basic domain and HA sequence.

3) Deletion in the Leucine Zipper domain
-5'PCR product
F- ACTAGTTGAAGTCGAAATATCCAGCACAGCC (*Spe*I)
R- ACTAGTCGTGTCCTCGGCGCGCTTCT (*SRS: Spe*I)
-3'PCR product
F- ACTAGTATGGAGCGCCAGCGGCAGAA (*SRS: Spe*I)
R- CCTAGGAAGAACGTTATAGAACCCATTCAAA (*Avr*II)
Exon7 alignment

|  |  |
|---|---|
| *Xrp1* | TTCGAGGCTCCCACCACAAGTACCAGTACCAGCACCAGCAACACTTCCATCAGCAGCAGC |
| *Xrp1*<sup>ΔLZHA</sup> | TTCGAGGCTCCCACCACAAGTACCAGTACCAGCACCAGCAACACTTCCATCAGCAGCAGC |
|  | `************************************************************` |
| *Xrp1* | ACCCATGCAGACATCGTAGAGGATCTACGCAGTGCCGAAGAGGAGACTACCACCGACTTC |
| *Xrp1*<sup>ΔLZHA</sup> | ACCCATGCAGACATCGTAGAGGATCTACGCAGTGCCGAAGAGGAGACTACCACCGACTTC |
|  | `************************************************************` |
| *Xrp1* | TCGGCGCCGAACACACCACACAGCAACTACTCAGCCAGCTCCAGCTGTGCGGCGCCCACC |
| *Xrp1*<sup>ΔLZHA</sup> | TCGGCGCCGAACACACCACACAGCAACTACTCAGCCAGCTCCAGCTGTGCGGCGCCCACC |
|  | `************************************************************` |
| *Xrp1* | TGCCAGACCGGCTATGGTGGCTTCCTCACTGCTCCCACCTCGCCCGCCTATTCAACAGCC |
| *Xrp1*<sup>ΔLZHA</sup> | TGCCAGACCGGCTATGGTGGCTTCCTCACTGCTCCCACCTCGCCCGCCTATTCAACAGCC |
|  | `************************************************************` |
| *Xrp1* | AGTACGTCCGTGTTCAGCCCTTCGCCAGCCAGCGGA**ATCAGCGGCAAGCGGAAGCGCGGT** |
| *Xrp1*<sup>ΔLZHA</sup> | AGTACGTCCGTGTTCAGCCCTTCGCCAGCCAGCGGAATCAGCGGCAAGCGGAAGCGCGGT |
|  | `************************************************************` |
| *Xrp1* | **CGCCCTGCCAAGGATCAT**GCCGACGGTCCCGATCCCGTGCTCATGTCAAGCATGAAAAGC |
| *Xrp1*<sup>ΔLZHA</sup> | CGCCCTGCCAAGGATCATGCCGACGGTCCCGATCCCGTGCTCATGTCAAGCATGAAAAGC |
|  | `************************************************************` |
| *Xrp1* | GAGGAGGAGAGGAAGGCCTACCAG*GACAGACTCAAGAACAACGAAGCGAGCCGCGTATCG* |

*Continued on next page*

| | |
|---|---|
| *Xrp1*<sup>ΔLZHA</sup> | GAGGAGGAGAGGAAGGCCTACCAGGACAGACTCAAGAACAACGAAGCGAGCCGCGTATCG |
| | \*\*\*\*\*\*\*\*\*\*\*\*\*\*\*\*\*\*\*\*\*\*\*\*\*\*\*\*\*\*\*\*\*\*\*\*\*\*\*\*\*\*\*\*\*\*\*\*\*\*\*\*\*\*\*\*\*\*\*\* |
| *Xrp1* | *CGCCGGAAGACGAAGGTGCGCGAAGAGGAGGAGAAGCGC***GCCGAGGACACGCTATTGGCC** |
| *Xrp1*<sup>ΔLZHA</sup> | CGCCGGAAGACGAAGGTGCGCGAAGAGGAGGAGAAGCGCGCCGAGGACACG--------- |
| | \*\*\*\*\*\*\*\*\*\*\*\*\*\*\*\*\*\*\*\*\*\*\*\*\*\*\*\*\*\*\*\*\*\*\*\*\*\*\*\*\*\*\*\*\*\*\*\*\*\*\*\* |
| *Xrp1* | **GAGAATCTGCGACTGCGTGCCCGGGCCGACGAAGTGGCCTCCCGGGAACGGAAGTTCAAG** |
| *Xrp1*<sup>ΔLZHA</sup> | -------------------------***ACTAGT***------------------------- |
| *Xrp1* | **AAGTACCTGATGGAGCGCCAGCGG**CAGAAGAGCACTTACGTTAAGCAGGAGCAG------ |
| *Xrp1*<sup>ΔLZHA</sup> | ---------ATGGAGCGCCAGCGGCAGAAGAGCACTTACGTTAAGCAGGAGCAG**TACCCC** |
| *Xrp1* | --------------------GAC**TGA**------GATTCTCGTAGCCATAGAATAGAAATT |
| *Xrp1*<sup>ΔLZHA</sup> | **TACGACGTCCCCGACTACGCG**GAC**TGA*GAATTC***GATTCTCGTAGCCATAGAATAGAAATT |
| | \*\*\*\*\*\*          \*\*\*\*\*\*\*\*\*\*\*\*\*\*\*\*\*\*\*\*\*\*\*\* |
| *Xrp1* | GCGGAGCATAAGCCTGTCATAGTACCTATCGTCGTCATCCAATTGTTTAGTTAGTTAAGT |
| *Xrp1*<sup>ΔLZHA</sup> | GCGGAGCATAAGCCTGTCATAGTACCTATCGTCGTCATCCAATTGTTTAGTTAGTTAAGT |
| | \*\*\*\*\*\*\*\*\*\*\*\*\*\*\*\*\*\*\*\*\*\*\*\*\*\*\*\*\*\*\*\*\*\*\*\*\*\*\*\*\*\*\*\*\*\*\*\*\*\*\*\*\*\*\*\*\*\*\*\* |
| *Xrp1* | TGTCGTTGAAGAAGTCGAGAAGCAGAAATCAAATTTCCATTTCCATGTGGACATCAGTTA |
| *Xrp1*<sup>ΔLZHA</sup> | TGTCGTTGAAGAAGTCGAGAAGCAGAAATCAAATTTCCATTTCCATGTGGACATCAGTTA |
| | \*\*\*\*\*\*\*\*\*\*\*\*\*\*\*\*\*\*\*\*\*\*\*\*\*\*\*\*\*\*\*\*\*\*\*\*\*\*\*\*\*\*\*\*\*\*\*\*\*\*\*\*\*\*\*\*\*\*\*\* |
| *Xrp1* | AGCCACTTATTAGGCATAGTGATGGCGCCCATAGTGCGCGAGGAGTGTTTAACTTTAATT |
| *Xrp1*<sup>ΔLZHA</sup> | AGCCACTTATTAGGCATAGTGATGGCGCCCATAGTGCGCGAGGAGTGTTTAACTTTAATT |
| | \*\*\*\*\*\*\*\*\*\*\*\*\*\*\*\*\*\*\*\*\*\*\*\*\*\*\*\*\*\*\*\*\*\*\*\*\*\*\*\*\*\*\*\*\*\*\*\*\*\*\*\*\*\*\*\*\*\*\*\* |
| *Xrp1* | TTACTTTTATTATGTTTGAGAAAACTAAAAAATTTTTATACCCACGGATATGTTTTTCGT |
| *Xrp1*<sup>ΔLZHA</sup> | TTACTTTTATTATGTTTGAGAAAACTAAAAAATTTTTATACCCACGGATATGTTTTTCGT |
| | \*\*\*\*\*\*\*\*\*\*\*\*\*\*\*\*\*\*\*\*\*\*\*\*\*\*\*\*\*\*\*\*\*\*\*\*\*\*\*\*\*\*\*\*\*\*\*\*\*\*\*\*\*\*\*\*\*\*\*\* |
| *Xrp1* | CCAAAGCGTAATCAAAAAAAAAAAAAGCACAACAAAAAAATGATTACGAGTTCGTATGTT |
| *Xrp1*<sup>ΔLZHA</sup> | CCAAAGCGTAATCAAAAAAAAAAAAAGCACAACAAAAAAATGATTACGAGTTCGTATGTT |
| | \*\*\*\*\*\*\*\*\*\*\*\*\*\*\*\*\*\*\*\*\*\*\*\*\*\*\*\*\*\*\*\*\*\*\*\*\*\*\*\*\*\*\*\*\*\*\*\*\*\*\*\*\*\*\*\*\*\*\*\* |
| *Xrp1* | TTAACCCCATTGATGAACAAAAAAAAGTTTAAAAAATGTCAACAAAAAAATATGCTATTG |
| *Xrp1*<sup>ΔLZHA</sup> | TTAACCCCATTGATGAACAAAAAAAAGTTTAAAAAATGTCAACAAAAAAATATGCTATTG |
| | \*\*\*\*\*\*\*\*\*\*\*\*\*\*\*\*\*\*\*\*\*\*\*\*\*\*\*\*\*\*\*\*\*\*\*\*\*\*\*\*\*\*\*\*\*\*\*\*\*\*\*\*\*\*\*\*\*\*\*\* |
| *Xrp1* | TTAAGAACAATTGAAATGTATATTTTACAGATATCAACTTATATAACCGTAATTGTTTGA |
| *Xrp1*<sup>ΔLZHA</sup> | TTAAGAACAATTGAAATGTATATTTTACAGATATCAACTTATATAACCGTAATTGTTTGA |
| | \*\*\*\*\*\*\*\*\*\*\*\*\*\*\*\*\*\*\*\*\*\*\*\*\*\*\*\*\*\*\*\*\*\*\*\*\*\*\*\*\*\*\*\*\*\*\*\*\*\*\*\*\*\*\*\*\*\*\*\* |
| *Xrp1* | ATAAAAAGCATGTTTCAATAAACGTGACAAAATCAACT |
| *Xrp1*<sup>ΔLZHA</sup> | ATAAAAAGCATGTTCAAATAAACGTGACAAAATCAACT |
| | \*\*\*\*\*\*\*\*\*\*\*\*\*\*\*\*\*\*\*\*\*\*\*\*\*\*\*\*\*\*\*\*\*\*\*\*\*\* |

The STOP codon is shown in bold black letters. Introduced restriction sites *Spe*I: ***ACTAGT*** and *Eco*RI: ***GAATTC*** are italicized in bold. Significant sequences: **AT-hook domain**, *Basic domain* (italicised), **Leucine zipper** and HA sequence.

Alignment of the C-terminal part of Xrp1 in several *Drosophila* species, showing the AT hook domain (with the crucial GRP tripeptide underlined) the Basic domain and the Leucine zipper. Additional underlined sequences in *D.melanogaster* indicate the extent of the deletions.

| | |
|---|---|
| *D.vir* | GFAGFITAPVSPAFSTASTSQFSVTTSITGNSSSKRKR<u>GRP</u>AKEHAEGPDPELMAQMTED |
| *D.pseudo* | GFGGFLTAPASPAYSVASTSQFSATTSTNSN-APKRKR<u>GRP</u>AKEHADGPDPEVMSRMDDE |

*Continued on next page*

| | |
|---|---|
| *D.erec* | GFGGFLTAPQSPAYSVASTSAFSPSPSG-GI-SGKRKRGRPAKDHADGPDPELMSGMTSE |
| *D.mel* | GYGGFLTAPTSPAYSTASTSVFSPSPAS-GI-SGKRKRGRPAKDHADGPDPVLMSSMKSE |
| *D.sim* | GYGGFLTAPASPAYSTASTSAFSPSPAG-GI-SGKRKRGRPAKDHADGPDPVLMSNMKSE |
| | *:.**:*** ***:*.**** ** : : . : *********:**:**** :*: * .: |
| *D.vir* | -DAKAYRDRIKNNEASRVSRRKTKQREHEEMKEEQDLQAEHEQLTYTLQLVMREARRYQE |
| *D.pseudo* | -KRKAYIDRIKNNEASRVSRRKTKSRDELEKQLEEELVAENERLLTQSQRVDHKETLFKN |
| *D.erec* | -EKKAYQDRIKNNEASRVSRRKTKKREEEEKSVEDELVAENLRLRTLADEVASKERKFKN |
| *D.mel* | EERKAYQDRLKNNEASRVSRRKTKVREEEEKRAEDTLLAENLRLRARADEVASRERKFKK |
| *D.sim* | EERKAYQDRLKNNEASRVSRRKTKVREEEEKRAEDELLAENLRLRAQADKVAFQERKFKK |
| | . *** **:************** *:. * *: * **. :* : * . ::: |
| *D.vir* | YLKRNYHKNSTYVKPEPDH |
| *D.pseudo* | YLMVRQRNNSTFVKKEH-- |
| *D.erec* | FLMLRMRKDSTFAEQE--- |
| *D.mel* | YLMERQRQKSTYVKQEQD |
| *D.sim* | YLMERQRMNSTYVKQEQD- |
| | :* . : .**:.: * |

## UAS-Xrp1<sup>ΔLZ-HA</sup> and UAS-Xrp1<sup>ΔBS-HA</sup>

The *Xrp1* long isoform cDNA (CDS2) was cloned into the vector pUAST as a *Bgl*II-*Not*I fragment (***Tsurui-Nishimura et al., 2013***). DNA fragments containing the modified C-terminal domains (*Xrp1*<sup>HA</sup>, *Xrp1*<sup>ΔBS-HA</sup> and *Xrp1*<sup>ΔLZ-HA</sup>) were cloned as *Stu*I-*Xmn*I fragments into pUAST-Xrp1 digested with *Stu*I and *Xba*I (made blunted). Similarly, a DNA fragment containing the deletion in the AT-Hook domain (*Xrp1*<sup>ΔAT-HA</sup>) was cloned as a *Bgl*I-*Xmn*I fragment into pUAST-Xrp1 digested with *Bgl*I and *Xba*I (made blunted).

The pUAST-Xrp1<sup>ΔC</sup> was generated by site-directed mutagenesis by introducing a single nucleotide change that converts Ser547 into a stop codon. The vector pUAST-Xrp1 was used as a template as well as the following primer combination (the introduced mutation is in red):

Xrp-Stop1-F CGCCCGCCTATTGAACAGCCAGTAC
Xrp-Stop1-R GTACTGGCTGTTCAATAGGCGGGCG

## Generation of the UAS-*Irbp18* and UAS-hDDIT3

The *irbp18* cDNA was amplified using as template a cDNA library obtained from third instar larvae total RNA. The following primer combination was used:

CTGTGTTTTCTTCTGCATTTCCGTC and
CACCAATAATGAAAACCATTTATGTAGGAT.

The human *DDIT3* cDNA was obtained from GenScript (catalog No: OHu16827D). Both cDNAs were sequenced and cloned into the pUAST vector. The resulting constructs (*pUAST-irbp18* and *pUAST-hDDIT3*) were injected into *Drosophila w*<sup>1118</sup> embryos following standard P-element transformation protocols (Rainbow Transgenics). More than five independent transgenic lines were established in each case.

## Acknowledgements

We thank D Rio for antibodies, N Phadnis for advice, M Kiparaki and J Secombe for comments on the manuscript, J DiGregorio for discussions, J Chuen for technical assistance and C Smadja for Blast searches. This work was supported by grants from the NIH (EY026720 and GM120451 to NEB, and Developmental Biology Training Grant 5T32 HD0741 to JCC). Drosophila stocks were obtained from the Bloomington Drosophila Stock Center (supported by NIH P40OD018537). Confocal microscopy was performed in the Analytical Imaging Facility of the Albert Einstein College of Medicine (supported by the NCI P30CA013330) using Leica SP5 and SP8 microscopes, the latter acquired through

NIH SIG 1S10 OD023591. DNA sequencing was performed by the Genomics Core of Albert Einstein College of Medicine.

## Additional information

### Funding

| Funder | Grant reference number | Author |
|---|---|---|
| National Eye Institute | EY026720 | Nicholas E Baker |
| National Institute of General Medical Sciences | GM120451 | Nicholas E Baker |
| Eunice Kennedy Shriver National Institute of Child Health and Human Development | 5T32 HD0741 | Jacob C Cooper |

The funders had no role in study design, data collection and interpretation, or the decision to submit the work for publication.

### Author contributions

Jorge Blanco, Conceptualization, Data curation, Formal analysis, Investigation, Methodology, Writing—original draft, Writing—review and editing; Jacob C Cooper, Conceptualization, Data curation, Formal analysis, Investigation, Methodology, Writing—review and editing; Nicholas E Baker, Conceptualization, Data curation, Funding acquisition, Investigation, Project administration, Writing—review and editing

### Author ORCIDs

Nicholas E Baker (iD) https://orcid.org/0000-0002-4250-3488

### Decision letter and Author response

Decision letter https://doi.org/10.7554/eLife.50535.sa1
Author response https://doi.org/10.7554/eLife.50535.sa2

## Additional files

### Supplementary files

- Supplementary file 1. Key Resources.
- Transparent reporting form

### Data availability

All data generated or analysed during this study are included in the manuscript and supporting files.

The following previously published dataset was used:

| Author(s) | Year | Dataset title | Dataset URL | Database and Identifier |
|---|---|---|---|---|
| Chang-Hyun Lee, Marianthi Kiparaki, Jorge Blanco, Virginia Folgado, Zhejun Ji, Amit Kumar, Gerard Rimesso, Nicholas E Baker | 2018 | RNA-seq analysis to assess transcriptional effects of Rp mutations in wing imaginal discs and their dependence on Xrp1 | https://www.ncbi.nlm.nih.gov/geo/query/acc.cgi?acc=gse112864 | GEO, GSE112864 |

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
