## [Decision Letter]

**Acceptance summary**

Based on the authors’ previous work, Minute cells reduce the expression of a number of genes that function in translation and Xrp1 has been found to function in a mechanism where the slow growth of one organ in the larva can result in a compensatory slowing of other organs so as to better synchronize their development. Here, the authors present compelling evidence that Irbp 18 functions in a similar way. Both Xrp1 and Irbp 18 likely function in a complex, promote their own expression, and Xrp1 seems to be the limiting component. The data suggest that upregulation of Xrp1 could be followed by an amplification step that may cause cell death. Both reviewers felt that the work is exciting and compelling.

**Decision letter after peer review**

Thank you for submitting your article "Roles of C/EBP class bZip proteins in the growth and cell competition of *Rp* ('Minute') mutants in *Drosophila*" for consideration by *eLife*. Your article has been reviewed by Utpal Banerjee as the Senior Editor, a Reviewing Editor (Hugo Bellen), and two reviewers. The following individuals involved in review of your submission have agreed to reveal their identity: Iswar K Hariharan (Reviewer #1).

The reviewers have discussed the reviews with one another and the Reviewing Editor has drafted this decision to help you prepare a revised submission.

I am enclosing both reviewers’ comments as both reviewers clearly find your work exciting and probing. They are both positive and one reviewer requests two experiments that are not very laborious and I would argue that they would be informative. I would suggest you carry out these experiments. If you feel strongly that they should not be carried out, please argue why.

*Reviewer #1:*

The phenomenon of cell competition was discovered in 1975 – clones of cells heterozygous for mutations in ribosomal proteins (Minutes) are eliminated when surrounded by wild-type cells. However, when tissues are composed entirely of those cells, they form normal adult structures albeit after a prolonged period of growth. Also, during cell competition, cell death occurs mainly in Minute clones where they abut wild-type cells. It was generally assumed that Minute cells are less fit because they have fewer ribosomes and that cell competition might be a mechanism of eliminating unfit cells.

Last year, the Baker lab published a very important paper where they showed that Minute cells do indeed have reduced rates of protein synthesis but not because they have fewer ribosomes. Rather, Minute cells increase the expression of a transcriptional regulator Xrp1, which reduces translation by reducing the expression of a number of genes that function in translation. Xrp1 has subsequently also been found to function in a mechanism where the slow growth of one organ in the larva can result in a compensatory slowing of other organs so as to better synchronize their developmental progression. Thus, the Xrp1 transcription factor is of great interest.

In this study, the Baker lab builds on those findings and further characterizes the role of Xrp1 in cell competition. First, they show that a likely dimerization partner, Irbp 18, functions, at least by genetic tests, in a similar way. They also show that a different dimerization partner of Irbp 18, ATf4 is unlikely to function in cell competition. Second, they show that both Xrp1 and Irbp 18, likely functioning in a complex, promote their own expression, and that Xrp1 is the limiting component. This suggests that an initial upregulation of Xrp1 could be followed by an amplification step that could reach a crucial threshold causing cell death. Third, they show that the DNA binding domains are necessary for these functions. Finally, they show that Xrp1 is a rapidly evolving gene and that human orthologs could potentially exist.

Overall, this paper provides some important results which enhance our understanding of cell competition. I do think that this paper is worthy of *eLife* but would ask the authors to perform two relatively simple experiments before this paper is accepted.

1) The upregulation of Xrp1 seems to occur uniformly throughout the mutant clones. If upregulation of Xrp1 is supposed to be responsible for Minute clone elimination, it does not explain why cell death mostly occurs near the boundaries of Minute clones. This could either be because Xrp1 upregulation induces some kind of secondary comparison mechanism at the boundaries. Alternatively, that comparison mechanism could operate independently of Xrp1 in Minute clones and both signals might be necessary for the cell death that occurs at clone boundaries. The authors are in a position to distinguish between these mechanisms by making FLP-out Xrp1-overexpressing clones using Gal80ts such that the expression of Xrp1, is activated when the clones is sufficiently large. If cell death is only observed near the clone boundary, this would indicate that Xrp1 is sufficient to make cells "losers" in a cell competition scenario. If the death is more uniform, it would argue that Xrp1 upregulation together with some other boundary effect generated by the Minute condition would together be responsible for the way Minute clones are eliminated. The current experiments do not distinguish between these possibilities.

2) Does uniform overexpression of Xrp1 throughout the compartment cause cell elimination e.g. using hh-Gal4? Compartments that are entirely Minute can be generated. Is this also true for compartments that are composed entirely of Xrp1 overexpressing cells?

The argument that Xrp1 and Irbp 18 function as a heterodimer would be strengthened by showing that these two proteins interact physically. Has this been shown by others already? The introduction only refers to an in vitro FRET assay. What about in vivo? If so, this should be mentioned clearly in the text.

Some of the writing can be simplified. For example, I had to read the following sentence multiple times to understand it:

"The requirement for DNA binding domains is consistent with transcriptional regulation of cell competition targets but an argument might also be made that Xrp1 and Irbp18 could be their own major transcriptional targets and that cell competition could depend on non-transcriptional mechanisms that nevertheless depended on transcriptional auto regulation to reach threshold expression levels."

Figure 3E: The legend indicates that non-GFP cells are Xrp1 +/-. Can't they be either Xrp1 +/- or Xrp1 +/+.

*Reviewer #2:*

In previous studies by the Baker lab it had been shown that the process of cell competition which leads to the elimination of cells that are hemizygous for genes encoding ribosomal proteins relies on the bZIP protein Xrp1. In an Xrp1 mutant background cells lacking one copy of a ribosomal gene can grow into clones like wild type cells, i.e. cell competition is not effective. The conclusion that cell competition is not a direct consequence of reduced ribosome abundance or decreased translation was surprising and has some interesting implications for maintenance of tissue integrity, for example in situations of somatically occurring chromosomal abnormalities such as trisomies or large-scale chromosomal mutations. Thus, this function of Xrp1 may be relevant for developmental disease, cancer and other pathologies and thus of considerable general interest.

The current manuscript presents data that strongly support the interpretation that the *Drosophila* IRBP18 bZIP protein is required for Xrp1 function, presumably by formation of a heterodimer between the two proteins. Xrp1 and IRBP18 are shown to both be essential for elimination of tissue that is hemizygous for ribosomal protein genes, that they are required for each other's up regulation in clones of heterozygous cells and that they prevent growth of resulting clones in a cell autonomous manner. Further, it is shown that the function of the presumed Xrp1/IRBP18 heterodimer is independent of Crc/ATF4, another bZIP protein that had been identified as a potential IRBP18 dimerization partner.

The presented conclusions are well supported by extensive and technically immaculate experiments.

The finding that Xrp1 requires a dimerization partner for its function is not particularly surprising, considering that it is a bZIP protein. Still, this is important information that extends our molecular understanding of a pathway that maintains genomic integrity of *Drosophila* and possibly other multicellular organisms.

The general relevance of this information is further supported by the last section of the paper in which potential functional homologs of Xrp1/IRBP18 in organisms other than dipterans are identified. Up to now it had been unclear whether such homologs exist. Baker et al. provide evidence that the Xrp1/ IRBP18 function is indeed conserved up to vertebrates, but that homologs are difficult to discern, presumably because the genes are undergoing rapid evolutionary change. The demonstration that a human C/EBP family member can functionally cooperate with Xrp1/ IRBP18 in cell competition when expressed in the fly supports the notion that transcriptional control of cell competition is an evolutionarily conserved process.

I recommend accepting this paper for publication.

---

## [Author Response]

Reviewer #1:1) The upregulation of Xrp1 seems to occur uniformly throughout the mutant clones. If upregulation of Xrp1 is supposed to be responsible for Minute clone elimination, it does not explain why cell death mostly occurs near the boundaries of Minute clones. This could either be because Xrp1 upregulation induces some kind of secondary comparison mechanism at the boundaries. Alternatively, that comparison mechanism could operate independently of Xrp1 in Minute clones and both signals might be necessary for the cell death that occurs at clone boundaries. The authors are in a position to distinguish between these mechanisms by making FLP-out Xrp1-overexpressing clones using Gal80ts such that the expression of Xrp1, is activated when the clones is sufficiently large. If cell death is only observed near the clone boundary, this would indicate that Xrp1 is sufficient to make cells "losers" in a cell competition scenario. If the death is more uniform, it would argue that Xrp1 upregulation together with some other boundary effect generated by the Minute condition would together be responsible for the way Minute clones are eliminated. The current experiments do not distinguish between these possibilities.

There are several reasons we prefer not to include these studies.

First, we probably cannot complete this experiment in the timeframe. Gal80[ts] would be necessary to control Xrp1 expression temporally, which would require culture at 18°C, which is slow.

Secondly, we doubt that results would be so simply interpretable. Xrp1 is extremely potent at killing cells (see response to the second experiment suggested) and finding an expression level that would mimic physiological levels to reveal a ‘boundary effect’ is likely to require some trial and error, if it is possible. It should be noted that Kocinski et al., (2017) similarly found, when activating the oxidative stress response, that ectopic activation was generally lethal and only very particular expression levels led to a boundary effect. Incidentally we have previously shown that Xrp1 is responsible for the oxidative stress response in Minute cells (Lee et al., 2018), which already suggests that Xrp1 should produce a boundary effect, through the oxidative stress response.

The alternative hypothesis suggested by the reviewer, which is that Xrp1 expression requires an independent signal in Minute cells for their death, might be true at physiological expression levels but we have already shown it cannot be true in ectopic expression studies because we and others have shown that Xrp1 misexpression kills non-Minute cells (e.g. Figure 5B). If the possibilities were as simple as the reviewer suggests, this experiment would already differentiate them.

2) Does uniform overexpression of Xrp1 throughout the compartment cause cell elimination e.g. using hh-Gal4? Compartments that are entirely Minute can be generated. Is this also true for compartments that are composed entirely of Xrp1 overexpressing cells?

We have expressed Xrp1 using en-G4 which but that genotype is embryonic lethal. Xrp1 is extremely potent at killing cells, e.g. in our hands it is more effective than ectopic expression of direct apoptosis components such as reaper or hid. This is why we were only able to perform structure-function studies with very late acting and tissue-specific Gal4 drivers such as GMR and Nubbin, and is another reason why we think the conditional clone approach suggested as experiment 1 will prove challenging. We now mention the enGal4 result and also explain why GMR and Nubbin drivers were used for ectopic expression studies (subsection “*Xrp1* domains necessary for ectopic function”)

The argument that Xrp1 and Irbp 18 function as a heterodimer would be strengthened by showing that these two proteins interact physically. Has this been shown by others already? The introduction only refers to an in vitro FRET assay. What about in vivo? If so, this should be mentioned clearly in the text.

We include references to the physical interaction of Irbp18 and Xrp1 proteins both purified from bacteria and in cultured cells (Discussion section).

Some of the writing can be simplified. For example, I had to read the following sentence multiple times to understand it:"The requirement for DNA binding domains is consistent with transcriptional regulation of cell competition targets but an argument might also be made that Xrp1 and Irbp18 could be their own major transcriptional targets and that cell competition could depend on non-transcriptional mechanisms that nevertheless depended on transcriptional auto regulation to reach threshold expression levels."

The problem sentence has been rewritten (subsection “*Irbp18* is required for ectopic *Xrp1* effects in wings and eyes”).

Figure 3E: The legend indicates that non-GFP cells are Xrp1 +/-. Can't they be either Xrp1 +/- or Xrp1 +/+.

The Figure 3E legend was revised to include this detail.